# QUALITY EVALUATION OF GANS USING CROSS LOCAL INTRINSIC DIMENSIONALITY

## ABSTRACT

Generative Adversarial Networks (GANs) are an elegant mechanism for data generation. However, a key challenge when using GANs is how to best measure their ability to generate realistic data. In this paper, we demonstrate that an intrinsic dimensional characterization of the data space learned by a GAN model leads to an effective evaluation metric for GAN quality. In particular, we propose a new evaluation measure, CrossLID, that assesses the local intrinsic dimensionality (LID) of input data with respect to neighborhoods within GAN-generated samples. In experiments on 3 benchmark image datasets, we compare our proposed measure to several state-of-the-art evaluation metrics. Our experiments show that CrossLID is strongly correlated with sample quality, is sensitive to mode collapse, is robust to small-scale noise and image transformations, and can be applied in a model-free manner. Furthermore, we show how CrossLID can be used within the GAN training process to improve generation quality.

## 1    INTRODUCTION

Generative Adversarial Networks (GANs) are powerful models for data generation, composed of two neural networks, known as the 'generator' and the 'discriminator'. The generator maps random noise vectors to locations in the data domain in an attempt to approximate the distribution of the input data. The discriminator accepts a data sample and returns a decision as to whether or not the sample is from the input or was artificially generated. While the discriminator is trained to distinguish input samples from generated ones, the generator's objective is to deceive the discriminator by producing data that cannot be distinguished from input data. The two networks are jointly trained to optimize an objective function resembling a two-player minimax game.

GANs were first formulated by Goodfellow et al. (2014), and have been applied to tasks such as image generation (Denton et al., 2015; Radford et al., 2016) and image inpainting (Pathak et al., 2016). Despite their elegant theoretical formulation (Goodfellow et al., 2014), training of GANs can be difficult in practice due to instability issues such as vanishing gradients and mode collapse. The vanishing gradient problem occurs whenever gradients become too small to allow sufficient progress towards an optimization goal within the allotted number of training iterations. The latter occurs when the generator produces samples for only a limited number of data modes, without covering the full distribution of the input data.

Deployment of GANs is further complicated by the difficulty of evaluating the quality of their output. Researchers often rely on visual inspection of generated samples, which is both time-consuming and subjective. A quantitative quality metric is clearly desirable, and several such methods do exist (Goodfellow et al., 2014; Salimans et al., 2016; Odena et al., 2017; Che et al., 2017; Heusel et al., 2017; Karras et al., 2018; Lucic et al., 2017; Lopez-Paz & Oquab, 2017; Shmelkov et al., 2018). However, past research has identified various limitations of some existing metrics (Theis et al., 2016; Barratt & Sharma, 2018), and effective evaluation of GAN models is still an open issue.

In this paper, we show how the data space learned by a GAN model can be understood in terms of the *Local Intrinsic Dimensionality* (LID) model of distance distributions (Houle, 2013). LID assesses the number of latent variables (the intrinsic dimensionality) needed to characterize the distribution of distances to a reference point $x$ — or equivalently, the discriminability of a distance measure in the vicinity of $x$. For the GAN discriminator, input samples are more likely to be discriminable from generated samples if the learning process maps them into a space where the distance measure

becomes more discriminable — or equivalently, one where the local intrinsic dimensionality is relatively low. For the GAN generator, a generated sample is more likely to be accepted as realistic if the learning process maps it into a local submanifold whose dimensionality matches that of its neighbors among the input data samples (after the mapping to the learned space). Our objective here is to develop this intuition into a technique for assessing the quality of the GAN learning process.

The main contributions of the paper are as follows:

- We propose CrossLID, a cross estimation technique based on LID that is capable of assessing the alignment of the data embedding learned by the GAN generator with that learned by the GAN discriminator.
- We show how CrossLID can be employed to avoid mode collapse during GAN training, to identify classes for which some or all modes are not well-covered by the learning process. We also show how this knowledge can then be used to bias the GAN discriminator via an oversampling strategy so as to improve its performance on such classes.
- Experimentation showing that our CrossLID measure is well correlated with GAN sample quality, and comparison to two state-of-the-art evaluation measures.

## 2 EVALUATION METRICS FOR GAN MODELS

GAN-based learning is an extensively researched area. Here, we briefly review the topic most relevant to our work, evaluation metrics for GAN models. Past research has employed several different metrics, including log-likelihood measures (Goodfellow et al., 2014), the Inception score (Salimans et al., 2016), the MODE score (Che et al., 2017), Kernel MMD (Gretton et al., 2006), the MS-SSIM index (Odena et al., 2017), the Fréchet Inception Distance (Heusel et al., 2017), the sliced Wasserstein distance (Karras et al., 2018), and Classifier Two-Sample Tests (Lopez-Paz & Oquab, 2017). In our study, we focus on the two most widely used metrics for image data, the Inception score (IS) and the Fréchet Inception Distance (FID), as well as a recently proposed measure, the Geometry Score (GS) (Khrulkov & Oseledets, 2018).

The Inception score uses an associated Inception classifier (Szegedy et al., 2016) to extract output class probabilities for each image, and then computes the Kullback-Leibler (KL) divergence of these probabilities with respect to the marginal probabilities of all classes:

$$\text{IS} = \exp(\mathbb{E}_{x \sim p_G} D_{\text{KL}}(p(y|x)||p(y)), \tag{1}$$

where $x \sim p_G$ implies a sample $x$ drawn from the generator outputs, $p(y|x)$ is the probability of class $y$ being assigned to $x$ by the Inception classifier, $p(y) = \int_x (p(y|x) dx$ is the marginal class distribution, and $D_{\text{KL}}$ is the KL divergence. IS measures two aspects of a generative model: 1) the images generated should be both clear and highly distinguishable by the classifier, as indicated by low entropy of $p(y|x)$ when marginalized over $y$, and 2) all classes should have good representation over the set of generated images, which can be indicated by high entropy of $p(y|x)$ when marginalized over $x$. However, a recent study has shown that IS is susceptible to variations in the Inception network weights when trained on different platforms (Barratt & Sharma, 2018).

The Fréchet Inception Distance (FID) passes both input and generated images to an Inception classifier, and extracts activations from an intermediate pooling layer. Activations are assumed to follow multidimensional Gaussians parameterized by their means and covariances. The FID is defined as

$$\text{FID} = (||\mu_I - \mu_G||)_2^2 + \text{Tr}(\Sigma_I + \Sigma_G - 2((\Sigma_I \Sigma_G)^{\frac{1}{2}})), \tag{2}$$

where $(\mu_I, \Sigma_I)$ and $(\mu_G, \Sigma_G)$ represent the mean and covariance of activations for input and generated data samples, respectively. Compared to IS, FID has been shown to more consistent with human judgment and more robust to noise; however, it also requires an external Inception classifier for its calculation (Heusel et al., 2017).

A recently proposed metric, the Geometry Score (GS) (Khrulkov & Oseledets, 2018) assesses the conformity between manifolds of input and generated data, in terms of the persistence of certain topological properties in a manifold approximation process. The topological relationships are extracted in terms of the counts of 1-dimensional loops in a graph structure built up from proximity relationships as a distance threshold is increased. Although it may be indirectly sensitive to variations

in the dimensionality of the manifolds, the Geometry Score (GS) explicitly rewards only matches in terms of the specific topology of these loop structures in approximations of the manifolds. However, due to its strictly topological nature, GS is insensitive to differences in relative embedding distances or orientations within the manifold — this issue is acknowledged by the authors, who advise that GS would be best suited for use in conjunction with other metrics (Khrulkov & Oseledets, 2018).

## 3 LOCAL INTRINSIC DIMENSIONALITY

In its most general sense, the Local Intrinsic Dimensionality (LID) is a characteristic of smooth functions that vanish at zero. The LID value can be regarded as the degree of the polynomial with the best fit to the function, taken over an infinitesimally small domain that includes the origin.

**Definition of LID:** To formalize LID, let $F$ be a function that is positive and continuously differentiable over some open interval containing $r > 0$. The LID of $F$ at $r$ is defined as:

$$\mathrm{LID}_F(r) := r \frac{F'(r)}{F(r)} = \lim_{\epsilon \to 0^+} \frac{\log\left(F((1+\epsilon)r)/F(r)\right)}{\log\left(1+\epsilon\right)} = \lim_{\epsilon \to 0^+} \frac{F((1+\epsilon)r) - F(r)}{\epsilon\, F(r)}, \qquad (3)$$

wherever the limits exists. The local intrinsic dimensionality of $F$ is then:

$$\mathrm{LID}_F^* = \lim_{r \to 0^+} \mathrm{LID}_F(r). \qquad (4)$$

In our context, and as originally proposed in (Houle, 2013), we are interested in functions that are the distributions of distances induced by some global distribution of data points: for each data sample generated with respect to the global distribution, its distance to a predetermined reference point determines a sample from the local distance distribution.

The LID model has the interesting property that the definition can be motivated in two different ways. The first limit stated in the definition follows from a modeling of the growth of probability measure in a small expanding neighborhood of the origin: as the radius $r$ increases, the amount of data encountered can be expected to grow proportionally to the $r$ to the power of the intrinsic dimension. Although the LID model is oblivious of the representational dimension of the data domain, in the setting of a uniform distribution with a manifold of dimension $m$, if $F$ is the distribution of distances to a reference point in the relative interior of the manifold, then $\mathrm{LID}_F^* = m$.

The second limit expresses the (in)discriminability of $F$ when interpreted as a distance measure evaluated at distance $r$ (with low values of $\mathrm{LID}_F(r)$ indicating higher discriminability). As implied by Eq. 3, the LID framework is extremely convenient in that the local intrinsic dimensionality and the discriminability of distance measures are shown to be equivalent and interchangeable concepts. For more information on the formal definition of LID and its properties, see (Houle, 2017a;b).

**Estimating LID:** LID is a generalization of pre-existing expansion-based measures which implicitly use neighborhood set sizes as a proxy for probability measure. These earlier models include the expansion dimension (Karger & Ruhl, 2002) and its variants (Houle et al., 2012a), and the minimum neighbor distance (MinD) (Rozza et al., 2012), all of which have been shown to be crude estimators of LID (Amsaleg et al., 2018). Although the popular estimator due to Levina & Bickel (2005) can be regarded as a smoothed version of LID, its derivation depends on the assumption that the observed data can be treated as a homogeneous Poisson process. However, the only assumptions made by the LID model is that the underlying (distribution) function be continuously differentiable.

For this work, we estimate LID using the Maximum Likelihood Estimator (MLE) as proposed in (Amsaleg et al., 2018), due to its ease of implementation and its superior convergence properties relative to the other estimators studied there. Given a set of data points $X$, and a distinguished data sample $x$, the MLE estimator of LID is:

$$\mathrm{LID}\,(x; X) = -\left(\frac{1}{k}\sum_{i=1}^{k} \ln \frac{r_i(x; X)}{r_{max}(x; X)}\right)^{-1} = \left(\ln r_{max}(x; X) - \frac{1}{k}\sum_{i=1}^{k} \ln r_i(x; X)\right)^{-1}. \qquad (5)$$

where $k$ is the neighborhood size, $r_i(x; X)$ is the distance from $x$ to its $i$-th nearest neighbor in $X \setminus \{x\}$, and $r_{\max}(x; X)$ denotes the maximum distance within the neighborhood (which by convention can be $r_k(x; X)$). Due to the deep equivalence between the LID model and the statistical theory of extreme values (EVT) shown in (Houle, 2017a; Amsaleg et al., 2018), the first of the two equivalent

formulations in Eq. 5 coincides with the well-known Hill estimator of scale derived from EVT (Hill, 1975). As can be seen from the second formulation, the reciprocal of the MLE estimator assesses the discriminability within the $k$-NN set of $x$ as the difference between the maximum and mean of log-distance values. Note that in these estimators, no explicit knowledge of the underlying function $F$ is needed - this information is implicit in the distribution of neighbor distances themselves.

Estimation of LID characteristics has seen practical applications for assessing the complexity of search queries in approximate similarity search (Houle et al., 2012b; Houle & Nett, 2015; Casanova et al., 2017), as a measure of the outlierness of data (Houle et al., 2018) and in the detection of adversarial samples for deep neural networks (Ma et al., 2018a). LID has also been used to detect and prevent overfitting by DNN classifiers on datasets with noisy labels (Ma et al., 2018b).

LID can characterize the intrinsic dimensionality of the data submanifold in the vicinity of a distinguished point $x$. The $\text{LID}(x; X)$ values of all data samples $x$ from a dataset $X$ can thus be averaged to characterize the overall intrinsic dimensionality of the manifold within which $X$ resides. In Romano et al. (2016); Houle (2017b), it was shown that this type of average is in fact an estimator of the correlation dimension over the sample domain (or manifold). Henceforth, whenever the context set $X$ is understood, we will use the simplified notation $\text{LID}(x)$ to refer to $\text{LID}(x; X)$, and to denote the average of these estimates over all $x \in X$ by $\text{LID}(X)$.

## 4 EVALUATING GANS VIA CROSS LOCAL INTRINSIC DIMENSIONALITY

We propose a new measure, CrossLID, that evaluates the closeness of the underlying submanifolds of an input distribution $p_I$ and a GAN-generated distribution $p_G$, as derived from the profiles of distances from samples of one distribution to samples of the other distribution. Our intuition is that if two distributions are similar, then the distance profiles of a generated sample with respect to a neighborhood of input samples should conform with the profiles observed within the input sample, and vice versa. As an illustration of the possible relationships between a generated distribution and the original input distribution, Figure 1 shows four examples of how a GAN model could learn a bimodal Gaussian distribution. Decreasing CrossLID scores indicate an increasing conformity between the two-mode input data distribution and the generated data distribution.

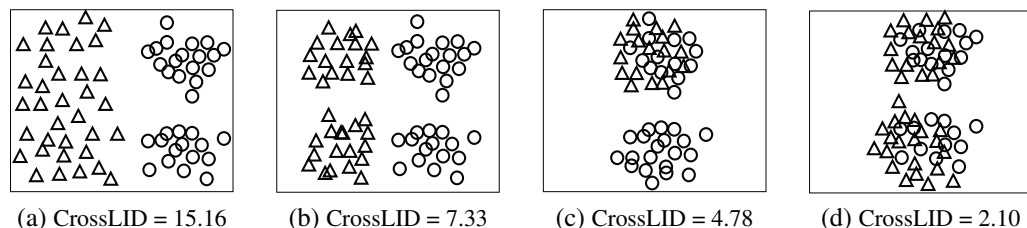

(a) CrossLID = 15.16     (b) CrossLID = 7.33     (c) CrossLID = 4.78     (d) CrossLID = 2.10

Figure 1: Four 2D examples showing how GAN-generated data samples (triangles) could relate to a bimodal Gaussian-distributed data set (circles), together with CrossLID scores: (a) generated data distributed uniformly, spatially far from the input data; (b) generated data with two modes, spatially far from the input data; (c) generated data associated with only one mode of the input data; and (d) generated data associated with both modes of the input data (the desired situation).

### 4.1 CROSSLID FOR GAN MODEL EVALUATION

We generalize the single data distribution based LID metric defined in Eq. 3 to a new metric that measures the cross LID characteristics between two distributions. Given two sets of samples $A$ and $B$, the CrossLID of samples in $A$ with respect to $B$ is defined as:

$$\text{CrossLID}(A; B) = \mathbb{E}_{x \in A}\text{LID}(x; B). \tag{6}$$

Note that $\text{CrossLID}(A; B)$ does not necessarily equal $\text{CrossLID}(B; A)$.

Low $\text{CrossLID}(A; B)$ scores indicate low average spatial distance between the elements of $A$ and their neighbor sets in $B$. From the second formulation of the LID estimator shown in Eq. 5, we see that increasing the separation between $A$ and $B$ would result in a reduction in the discriminability

of distances between them, as assessed by the difference between the maximum and mean of the log-distances from points of $A$ to their nearest neighbors in $B$ — thereby increasing the CrossLID score. As a simplified example, consider the situation in which a positive correction $d$ is added to each of the distances from some reference sample $x \in A$ with respect to its neighbors in $B$. This distance correction would cause the reciprocal of the LID estimate defined in Eq. 5 to become $\ln(r_{\max}+d) - \frac{1}{k} \sum_{i=1}^{k} \ln(r_i+d)$, which leads to an increase in the estimate of LID when $d > 0$, and a decrease when $d < 0$. Thus, a good alignment between $A$ and $B$ is revealed by good discriminability (low LID) of the distance distributions induced by one set ($B$) relative to the members of the other ($A$). In general, CrossLID differs from LID in its sensitivity to differences in spatial position and orientation of the respective manifolds within which $A$ and $B$ reside (see Appendix A).

Unlike methods based on thresholding of neighbor distances (such as Hausdorff distance or other linkage criteria from clustering), LID scores are naturally adaptive to local differences in intrinsic dimensionality. Although the two-dimensional point configurations shown in Figure 1 are amenable to such techniques, they quickly break down for data in higher dimensions, or across a range of different local intrinsic dimensionalities. Without taking local intrinsic dimensionality into account, we would not know whether a given large 1-NN distance value indicates 'large spatial separation' or 'conformity within a locality of high intrinsic dimensionality' — implying that the direct use of 1-NN distance information leads to a rather poor assessment of the relationship of a point to its surroundings. This observation is borne out by the evaluation in (Amsaleg et al., 2018) that an estimator of local intrinsic dimensionality using only the 1-NN and k-NN distance measurements (a variant in the 'MiND' family) was shown to lead to relatively poor performance.

Low values of CrossLID($A; B$) also indicate good coverage of the domain of $A$ by elements of $B$. To see why, consider what would happen if this were not the case: if the samples in $B$ did not provide good coverage of all modes of the underlying distribution of $A$, there would be a significant number of samples in $A$ whose distances to its nearest neighbors in $B$ would be excessively large in comparison to an alternative set $B'$ providing better coverage of $A$ (see Figure 1c and 1d for an example). As discussed above, this increase in the distance profile would likely lead to an increase in many of the individual LID estimates that contribute to the CrossLID score.

Given a set of samples $X_I$ from a GAN input distribution, and a set of samples $X_G$ from the GAN generated distribution, a low value of CrossLID($X_I; X_G$) indicates a good alignment between the manifold associated with $X_G$ and the manifold associated with $X_I$, as well as an avoidance of mode collapse in the generation of $X_G$. It should be noted, however, that low values of CrossLID($X_G; X_I$) do not discourage mode collapse. Since low values of CrossLID($X_I; X_G$) encourage a good integration of generated data into the submanifolds with respect to these learned representations, and an avoidance of mode collapse in sample generation, CrossLID($X_I; X_G$) is a good candidate measure for evaluating GAN learning processes.

CrossLID also allows targeted quality assessment of GANs for refined sample groups of interest. For example, for a specific mode ($X_I^m$) from the input samples based on either cluster information or class information, CrossLID($X_I^m; X_G$) can be used to assess how well the GAN model learns the submanifold of this particular mode. CrossLID can therefore be exploited to detect and mitigate underlearned modes in GAN training. We will explore this further in Section 5.

## 4.2 Effective Estimation of CrossLID

We next discuss two important aspects in CrossLID estimation: 1) the choice of feature space where CrossLID is computed and 2) the choice of appropriate sample and neighborhood sizes for accurate and efficient CrossLID estimation.

**Deep Feature Space for CrossLID Estimation:** The representations that define the underlying manifold of a data distribution are well learned in the deep representation space. Recent work in representation learning (Goodfellow et al., 2016), adversarial detection (Ma et al., 2018a) and noisy label learning (Ma et al., 2018b) has shown that DNNs can effectively map high-dimensional inputs to low-dimensional submanifolds at different intermediate layers of the network. We denote the output of such a layer as a function $f(x)$, and estimate CrossLID in the deep feature space as:

$$\text{CrossLID}(f(X_I); f(X_G)) = \frac{1}{|X_I|} \sum_{x \in X_I} \left( \ln r_{\max}(f(x), f(X_G)) - \frac{1}{k} \sum_{i=1}^{k} \ln r_i(f(x), f(X_G)) \right)^{-1}. \quad (7)$$

It should be noted that successful learning by the GAN discriminator would entail the learning of a mapping $f$ for which the intrinsic dimensionality of $f(X_I)$ is relatively low, and the local discriminability is relatively high. This encourages the GAN generator to produce samples for which $\text{CrossLID}(f(X_I); f(X_G))$ is also low, which further enhances the value of CrossLID in GAN evaluation and training.

Note that CrossLID can be computed using a single forward pass of an available deep neural network model — no backward pass is needed. The transformation $f(x)$ can be computed using an external network trained separately on the real data distribution, such as the Inception network used by Inception score and FID. It could also be the discriminator network of a GAN, as they are known to be capable of learning quality representations suitable for classification (Radford et al., 2016). In Section 6.1 we will show that both choices work well for the estimation of CrossLID.

**Sample Size and Neighborhood for CrossLID Estimation:** Searching for the $k$-nearest neighbors of all input samples of $X_I$ within the entire GAN-generated dataset $X_G$ can be prohibitively expensive. Although most input data sets require neighborhood sizes on the order of $k = 100$ for the convergence of the LID estimators (Amsaleg et al., 2018), previous work using the LID measure in adversarial detection (Ma et al., 2018a) and noisy label learning (Ma et al., 2018b) has demonstrated that LID estimation at the deep feature level can be effectively performed within small batches of training samples — with neighborhood sizes as small as $k = 20$ drawn from batches of 100 samples. For the estimation of $\text{CrossLID}(f(X_I); f(X_G))$, we use $|X_I| = 20000$ input samples and $|X_G| = 20000$ GAN-generated samples. For each input sample $f(x)$ where $x \in X_I$, we search $k = 100$ nearest neighbors within 1000 samples selected randomly from $f(X_G)$, and use the distances from $f(x)$ to these $k = 100$ nearest neighbors to estimate $\text{CrossLID}(f(x); f(X_G))$. The mean of the CrossLID estimates over all 20000 input samples determines the final overall estimate.

# 5 OVERSAMPLING IN GAN TRAINING WITH MODE-WISE CROSSLID

A GAN distribution may not equally capture the distributions of all modes presenting in a real data distribution. Due to the inherent randomness in stochastic learning, the decision boundary of the discriminator may be closer to regions of some modes than others at different stages of the training process. The closer modes may develop stronger gradients, in which case the generator would learn these modes better than the others. If imbalances in learning can be detected and addressed during training, we could expect a better convergence to good solutions. To achieve this, we propose a GAN training strategy with oversampling based on mode-wise CrossLID scores (as defined in section 4.1).

We describe our training strategy in the context of labeled data, where we simply take the classes to be the modes. Note that for unlabeled data, the modes can be determined by computing clusters within the data. Here, we compute the average CrossLID score for samples from each class, and use it to measure how well a class has been learned — the lower the CrossLID score, the more effective the learning. To help generate good gradients for all classes during the training, we dynamically modify the input of the discriminator by oversampling the poorly learned classes (those with high class-wise CrossLID scores). The objective is to bias the discriminator's decision boundary towards the regions of poorly learned classes in order to produce stronger gradients for the generator in favor of underlearned classes.

The steps are described in Algorithm 1. For each class $c \in \{1, \cdots, C\}$, we select a subset of input samples $X'_c$ from that class, of size proportional to a deviation factor $\gamma_c = (\text{CrossLID}(X_I^c; X_G) - \text{CrossLID}(X_I^c; X_I^c))/\text{CrossLID}(X_I^c; X_I^c)$, and augment the original input dataset with the members of $X'_c$ for subsequent training. $\gamma_c$ measures the relative deviation of the CrossLID score $\text{CrossLID}(X_I^c; X_G)$ from the self-CrossLID score $\text{CrossLID}(X_I^c; X_I^c)$, the correlation dimension of the class input data distribution. When the GAN model has already fully learned the distribution of a given class (that is, when $\text{CrossLID}(X_I^c; X_G) = \text{CrossLID}(X_I^c; X_I^c)$), $\gamma_c = 0$, indicating that no oversampling will be applied to this class.

Our proposed strategy can effectively deal with the mode collapse issues encountered in GAN training. When the generator learns a class only partially, or not at all, it will receive a relatively high CrossLID score for that class. In subsequent iterations, the imbalance in learning will be addressed by our oversampling step in favor of these classes (Step 8 in Algorithm 1).

---

**Algorithm 1** Oversampling in GAN training with mode-wise CrossLID

---

1: **for** every $T$ generator iterations **do**
2:     Generate $N_1$ GAN samples $X_G$.
3:     **for** $c$ in $\{1, \cdots, C\}$ **do**                                                   ▷ $C$: number of classes.
4:         Sample $N_2$ input samples $X_I^c$ from class $c$
5:         $\gamma_c = (\text{CrossLID}(X_I^c; X_G) - \text{CrossLID}(X_I^c; X_I^c))/\text{CrossLID}(X_I^c; X_I^c)$
6:     **end for**
7:     $\gamma_c = \gamma_c / \sum_{j=1}^{C} \gamma_j$, for $c \in \{1, \ldots, C\}$.              ▷ Normalization for next step oversampling.
8:     $X_{\text{aug}} = \{X_1', \cdots, X_C'\} \cup X_I$ where $X_c'$ is a random sample from $X_I^c$ of size $|X_c'| = m \times \gamma_c$
    and $m$ is a size parameter
9:     Continue GAN training with $X_{\text{aug}}$ for the next $T$ generator iterations.
10: **end for**

---

# 6   EXPERIMENTAL RESULTS

## 6.1   EVALUATION OF CROSSLID AS A GAN QUALITY METRIC

We first demonstrate that the CrossLID score is well correlated with sample quality of GANs. We then assess CrossLID with respect to the following considerations: 1) sensitivity to mode collapse, 2) robustness to input noise, 3) robustness to image transformations, 4) robustness to sample size used for estimation, and 5) dependency on external models. We also compare CrossLID score with Geometry score, Inception score and FID, on MNIST (LeCun et al., 1990), CIFAR-10 (Krizhevsky & Hinton, 2009) and SVHN (Netzer et al., 2011) datasets.

For the CrossLID score, we used external CNNs trained on the original training set of real images for feature extraction (see Appendix N.2 for more details), except for evaluation scenario 5. To compute the Inception score, we followed Salimans et al. (2016) using the pretrained Inception network, except in the case of MNIST, for which we pretrained a different CNN model as described in Li et al. (2017). FID scores were computed as in Heusel et al. (2017). Our code is available at `https://www.dropbox.com/s/bqadqzr5plc6xud/CrossLIDTestCode.zip`.

**Correlation of CrossLID and Sample Quality of GANs:** We show that the CrossLID score is strongly correlated with sample quality of GAN models. In the left three subfigures of Figure 2, as GAN training proceeds, the CrossLID score decreases. CrossLID$(X_I; X_G)$ was estimated over 20000 generated samples using deep features extracted from the external CNN model. The rightmost subfigure in Figure 2 illustrates the negative correlation between CrossLID score and Inception score over different training epochs. (Supporting images for visual verification of correlation between CrossLID and sample quality can be found in Appendix E.) We also found that the Geometry Score did not exhibit a clear correlation with sample quality, which is consistent with its reported insensitivity to differences in embedding distances or orientations (Khrulkov & Oseledets, 2018) (see Appendix C). For this reason, we omit the Geometry Score from the remainder of the discussion.

**Sensitivity to Mode Collapse:** A challenge of GAN training is to overcome mode collapse, which occurs when the generated samples cover only a limited number of modes (not necessarily from the real distribution) instead of learning the entire real data distribution. An effective evaluation metric for GANs should be sensitive to such situations.

We simulate two types of mode collapse by downsampling the training data: 1) intra-class mode dropping, which occurs when the GAN generates samples covering all classes, and 2) inter-class mode dropping, which occurs when the GAN generates samples from a limited number of classes. For both types, we randomly select a subset of $n$ samples from $c$ classes from the original training set (of $N$ samples from $C$ classes), then randomly subsample with replacement from the subset to create a new dataset with the same number of samples $N$ as in the original training set. For the simulation of intra-class mode dropping, we let $c = C$, and vary $n \in [30, 40, 50, 70, 100]$, whereas for inter-class mode dropping we let $n = 50$, and vary $c \in [2, 4, 6, 8, 10]$. Overall, for each of the original datasets, we created five new datasets for each type of mode collapse, and computed CrossLID, Inception, and FID scores on the new datasets. Note that each of these new datasets has the same number of instances $N$ as the original training set from which it was derived.

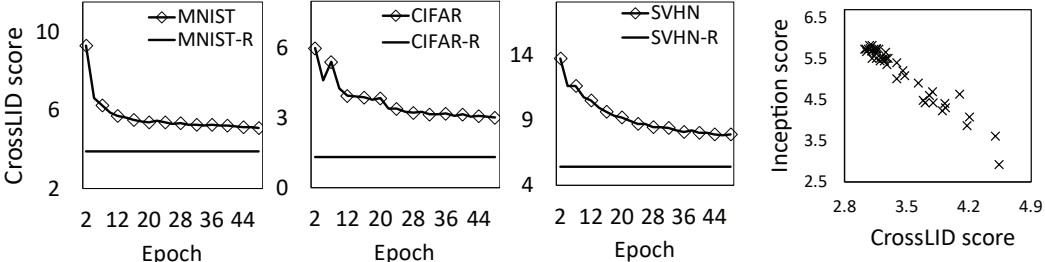

Figure 2: *Left three*: The CrossLID($X_I$; $X_G$) score for generated samples by a DCGAN after different training epochs. Results are shown for the first 50 epochs of training on MNIST, CIFAR-10 and SVHN, and MNIST-R/CIFAR-R/SVHN-R denote the CrossLID($X_I$; $X_I$) scores for real MNIST/CIFAR-10/SVHN samples. *Rightmost*: Correlation between CrossLID score and Inception score for CIFAR-10 dataset, each point is associated with a model at a certain epoch.

As shown in Figure 3, we found that CrossLID is sensitive to different degrees of intra-class mode dropping, but the Inception score failed to identify intra-class mode dropping on MNIST and SVHN, and responded inconsistently for different levels of intra-class mode dropping on CIFAR-10. FID is also sensitive to intra-class mode dropping. Similar results were seen for inter-class mode dropping (see Figure 8 in Appendix B): again, CrossLID was found to be sensitive to increasing levels of inter-class mode dropping, and is more sensitive than FID. Although the Inception score revealed inter-class mode dropping for MNIST, it failed to do so for CIFAR-10 and SVHN.

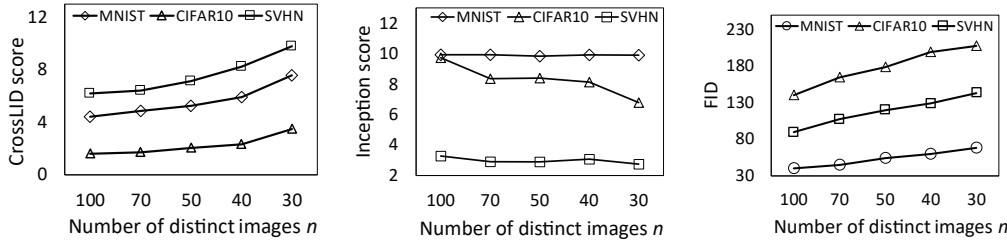

Figure 3: Test results for intra-class mode dropping: The CrossLID scores (left), Inception scores (middle) and FID scores (right) on varying numbers of unique samples in the datasets.

**Robustness to Small Input Noise:** We further examine the robustness of the three scores to small levels of Gaussian input noise. We add input noise drawn from a Gaussian distribution with both mean and variance equal to 127.5 (255/2) to a certain proportion of pixels in the original images. In Figure 4, CrossLID exhibited small variation as the proportion of modified pixels was increased from 0.2% to 2%. In contrast, the Inception score and FID both demonstrated large variations, particularly for CIFAR-10 and SVHN. For low levels of noise that do not substantially change the visual quality of the image, we believe that robustness is a desirable characteristic for a quality measure (for further discussion of this issue, see Appendices G and H). Noting that there is as yet no consensus on the issue of whether GAN quality measures should be robust to noise, we pose it as an open problem for the GAN research community to explore.

**Robustness to Input Transformation:** We test the robustness of the measures to small input transformations. As long as the transformations do not alter the visual appearance of GAN images, a robust metric should be able to give consistent evaluations. This is important in that GAN generated images often exhibit small distortions compared to natural images, and such small imperfections should not significantly detract from the perceived quality of GANs. However, as demonstrated in the left and middle subfigures of Figure 5, while CrossLID and Inception score are moderately robust to small translations and rotations on CIFAR-10 images, the FID score increased significantly as the distortion increased. FID calculation on different (non-Inception) feature spaces could possibly lead to different behavior; however, this investigation is beyond the scope of this paper.

**Robustness to Sample Size:** For the sake of efficiency, it is desirable that GAN quality measures be able to perform well even when computed over relatively small sample sizes. We test the robustness

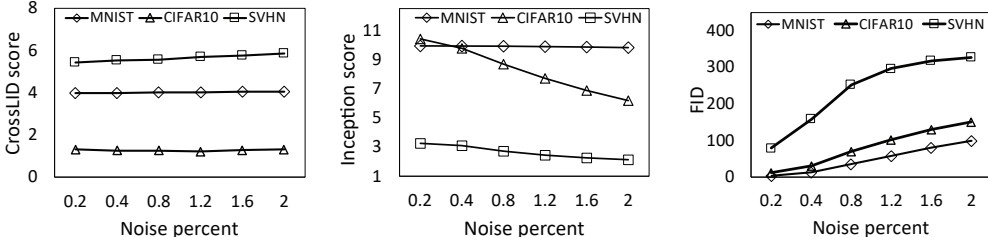

Figure 4: Robustness to Gaussian noise for CrossLID, Inception score and FID score (from left to right). Noise percent indicates proportion of pixels of GAN images that have been added noise.

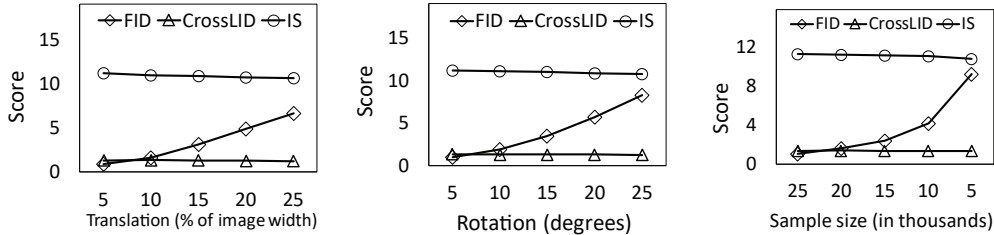

Figure 5: Robustness test of the three metrics on CIFAR-10 dataset to small image transformations including translation (left ) and rotation (middle), and sample size used for calculation (right).

of the three metrics versus sample size on a subset of CIFAR-10 training images. The results are shown in the right subfigure of Figure 5. CrossLID and the Inception score are stable as the subset size decreases from 25K to 5K. However, the FID score turns out to be quite sensitive to the sample size. The lower variation of CrossLID against sample size allows it to be computed on samples of smaller size as compared to what is typically needed by the other two metrics. We note that previous research on FID (Lucic et al., 2017) has noted that it exhibits high variance for low sample sizes, and has hence been recommended only for sufficiently large sample sizes ($> 10K$).

**Dependency on External Models:** So far we have evaluated the CrossLID score using a separately trained CNN classifier as the feature extractor for its estimation in deep feature space. Recent work has demonstrated that the discriminator network of a GAN model is capable of learning class-independent features that are suitable for classification (Radford et al., 2016). Here, we show that our proposed CrossLID score can be accurately estimated directly using the feature space of the discriminator network, and that an external feature extractor is not necessary. We compute CrossLID scores of different DCGANs (Radford et al., 2016) trained on MNIST, CIFAR-10, and SVHN, based on the discriminator's outputs at the second-to-last layer. We train all GANs for 50 epochs and report the CrossLID score. As shown in Fig 2, the CrossLID scores computed using the discriminator are correlated across training epochs with CrossLID scores computed using an external feature extractor. (the results are reported in Appendix K). This indicates that CrossLID may be computed without a pre-trained model, and therefore can be used to evaluate GANs trained on non-image data or unlabeled data. As with CrossLID, FID may also be computed in the feature space of discriminators and other models. However, the Inception score requires class labels, and is therefore dependent on a pre-trained classifier using labeled data.

**Summary:** Table 1 summarizes our experimental comparison of CrossLID, the Inception score, and FID as per its original formulation computed on the activations of the Inception model, rather than in other feature spaces such as those that could potentially be generated using discriminator models.

## 6.2 EVALUATION OF THE PROPOSED GAN TRAINING WITH OVERSAMPLING

We evaluate the effectiveness of the CrossLID-guided oversampling approach in GAN training. For the MNIST, CIFAR-10 and SVHN datasets, we compare standard versions of the popular DC-GAN (Radford et al., 2016) and WGAN (Arjovsky et al., 2017) models to the same models trained with CrossLID-guided oversampling. (Further details of model architecture, experimental settings, and output images can be found in Appendix N.1.)

Table 1: Evaluation results of CrossLID, Inception score and FID.

| EVALUATION CRITERIA | CrossLID score | Inception score | FID |
|---|---|---|---|
| Sensitivity to mode collapse. | High | Low | High |
| Sensitivity to small input noise. | Low | High | High |
| Sensitivity to small input transformations. | Moderate | Moderate | High |
| Sensitivity to sample size for estimation. | Low | Moderate | High |
| Dependency on external model. | Optional | Dependent | Optional |

The performances in terms of CrossLID scores are reported in Table 2, where DCGAN+ and WGAN+ refer to training with our proposed oversampling. Our training approach achieved better CrossLID scores than standard training for both DCGANs and WGANs. (Inception scores and FID results for these experiments are reported in Appendix L.)

Table 2: Performance of CrossLID-guided oversampling on DCGAN and WGAN.

| Dataset | CrossLID score (lower is better) | | | |
|---|---|---|---|---|
| | DCGAN | DCGAN+ | WGAN | WGAN+ |
| MNIST | $5.11 \pm 0.02$ | $\mathbf{4.96 \pm 0.08}$ | $5.91 \pm 0.02$ | $\mathbf{5.26 \pm 0.02}$ |
| CIFAR10 | $3.00 \pm 0.04$ | $\mathbf{2.78 \pm 0.04}$ | $3.70 \pm 0.04$ | $\mathbf{3.57 \pm 0.04}$ |
| SVHN | $7.40 \pm 0.01$ | $\mathbf{7.14 \pm 0.03}$ | $10.14 \pm 0.04$ | $\mathbf{9.95 \pm 0.04}$ |

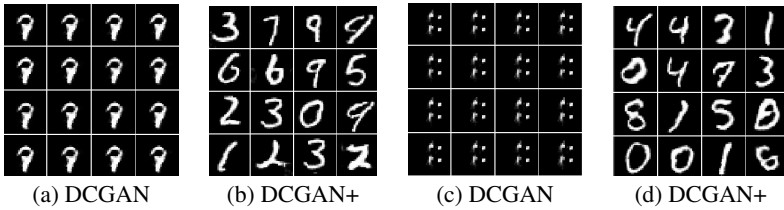

(a) DCGAN      (b) DCGAN+      (c) DCGAN      (d) DCGAN+

Figure 6: Images generated at the end of the 30-th epoch by DCGAN and DCGAN+ on the MNIST dataset, with (a-b) no batch normalization in the generator, and (c-d) no batch normalization in either the generator or the discriminator.

**Effectiveness in Preventing Mode Collapse:** As explained in section 5, our approach can help avoid mode collapse. We next show on MNIST, when the batch-normalization (BN) layers were removed from the generator, or from both the discriminator and the generator, standard DCGAN training suffered significant mode collapse and failed to learn the full real distribution, as shown in Figures 6 (a) and (c). Our approach, however, was still able to produce high quality images without any sign of mode collapse during the training, as shown in Figures 6 (b) and (d). (More details of the training process and visual inspections can be found in Appendix M.)

## 7 CONCLUSION

We have proposed a new metric for quality evaluation of GANs, based on cross local intrinsic dimensionality (CrossLID). Our measure can effectively assess sample quality and mode collapse in GAN outputs. It is reasonably robust to input noise, image transformations, and sample size. We also demonstrated a simple oversampling approach based on the mode-wise CrossLID that can improve GAN training and help avoid mode collapse.

We believe CrossLID is not only a promising new tool for assessing the quality of GANs, but also a promising tool to help improve GAN training strategies. We envisage CrossLID can be used as an additional metric for the community to evaluate GAN quality. Unlike Inception score and FID, CrossLID uses a local perspective rather than global perspective when evaluating sample quality, in that a quality score for each individual GAN generated sample can be computed based on its neighborhood. The advantage of mode-wise performance estimation by CrossLID may be utilized in different GAN models such as conditional and supervised GANs. Similar to current practice in clustering algorithm research, reporting multiple quality performance measures for a clustering (GAN distribution) is likely to be more robust than reporting just a single performance measure.

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

# A    THE EFFECT OF MANIFOLD POSITIONING AND ORIENTATION ON CROSSLID

In Figure 1 of Section 4, we have shown the power of CrossLID in capturing submanifold closeness via a toy example where a GAN model attempts to learn a bimodal Gaussian distribution. Here, we provide more insights into the understanding of CrossLID. Figure 7 illustrates how our proposed CrossLID can effectively characterize the closeness of two manifolds ($X$ and $Y$) with identical geometric structures but different positioning or orientation in space. As the two manifolds move closer and closer to each other either in position (decreasing distance $d$) or orientation (decreasing angle $\theta$), the CrossLID scores (CrossLID($X; Y$)) tend to decrease, and are close to one (CrossLID($X; X$)) when the two manifolds are completely overlapping. This substantiates how effectively CrossLID can measure differences in spatial position and orientation of two manifolds, even when they have similar structure.

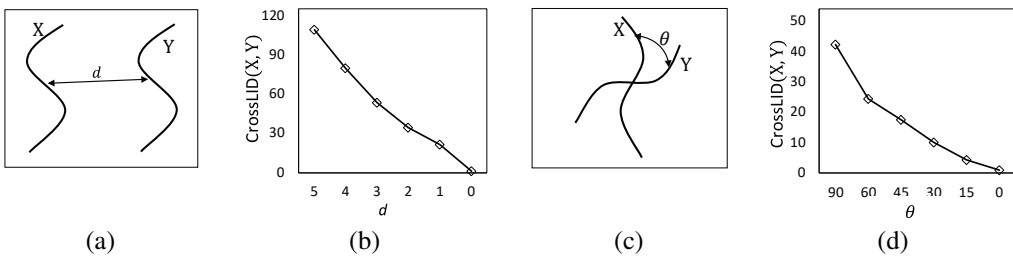

Figure 7: (a) Two manifolds X and Y have identical geometric structures, but different positions in space; (b) CrossLID score decreases as X and Y moving closer to each other (decreasing $d$); (c) The same manifolds (as in (a)) but having different orientations (rotated with respect to each other); (d) CrossLID score decreases as the relative orientation angle $\theta$ decreases.

# B    TEST RESULTS FOR INTER-CLASS MODE DROPPING

In Section 6.1 we discussed on the performance of CrossLID, FID, and the Inception score in the presence of intra- and inter-class mode dropping; the experimental results are quantified here, in Figure 8. CrossLID was found to be sensitive to increasing levels of inter-class mode dropping, and is more sensitive than FID. Although the Inception score revealed inter-class mode dropping for MNIST, it failed to do so for CIFAR-10 and SVHN.

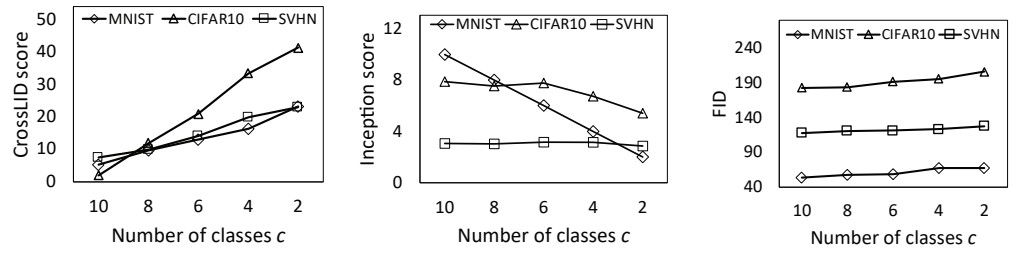

Figure 8: Test results for inter-class mode dropping: The CrossLID scores (left), Inception scores (middle) and FID scores (right) on varying numbers of unique classes in the datasets.

# C    SAMPLE QUALITY EVALUATION OF GEOMETRY SCORE

We train DCGANs on MNIST, CIFAR-10 and SVHN datasets, and compute the Geometry score using 2000 generated samples after each epoch of training for the first 50 epochs. The DCGAN architectures used here are the same as used in the other experiments of Section 6.1, and are described in Appendix N.1. As demonstrated in Figure 9, the Geometry Score exhibits high variation and has

no clear correlation with the sample quality, which is consistent with the claims in (Khrulkov & Oseledets, 2018). Visual verification of the improving sample quality over epochs can be found in Figures 11 and 12. Although a larger sample size (10000) was used in the original paper, we found it is computationally expensive to compute Geometry scores with such a large sample size over many epochs.

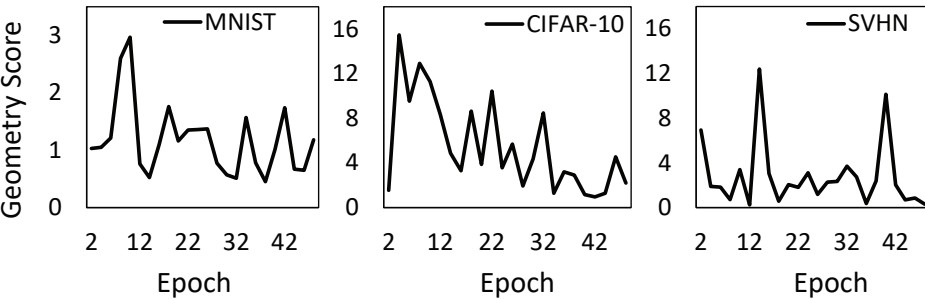

Figure 9: Evaluation of Geometry score on sample quality for DCGAN models trained on MNIST (left), CIFAR-10 (middle) and SVHN (right). After each epoch of training (for the first 50 epochs), we compute the Geometry Score over 2000 samples generated by the network.

## D    BENEFITS OF CROSSLID ESTIMATION IN DEEP FEATURE SPACE

In Section 4.2 we have proposed the use of deep feature space for the estimation of CrossLID. Following the discussion in Section 6.1, we show here experimental details justifying the avantages CrossLID over Inception score and FID when computed within deep feature spaces. We test the estimation of CrossLID on purely real samples (that is, CrossLID$(X_I, X_I)$) on the MNIST dataset for two settings: 1) directly in the pixel space, or 2) in deep feature space as defined by an external CNN classifier. Figure 10 demonstrates the robustness of CrossLID in the two settings for three scenarios: 1) small scale input noise, 2) translations, and 3) rotations. The CrossLID scores estimated in the feature space remains largely invariant across all test scenarios, whereas the scores estimated in the pixel space exhibit some variation (increases in value). This confirms the denoising and representation learning capabilities of convolutional networks, and the advantage of CrossLID estimation in deep feature space.

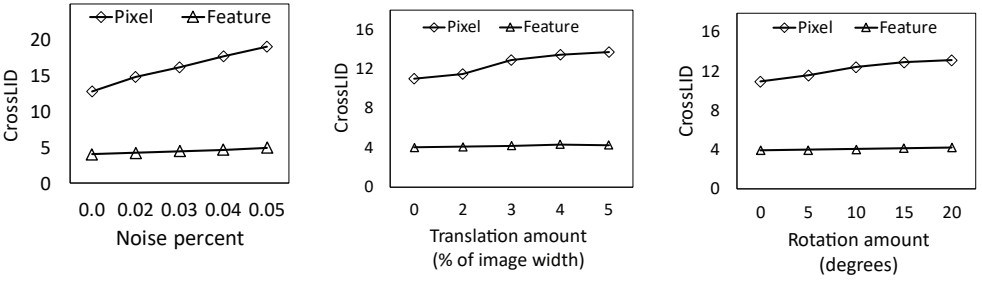

Figure 10: Comparison of CrossLID scores estimated in the pixel space versus those estimated in the deep feature space, on the MNIST dataset, against small-scale salt-and-pepper (impulse) noise (left), translations (middle) and rotations (right).

## E    VISUAL INSPECTIONS ON CROSSLID SCORE AND SAMPLE QUALITY

In Section 6.1, we demonstrated that the CrossLID score is closely correlated with the sample quality in a GAN training process, in that the CrossLID score decreases consistently as the model progressively learns to generate samples of better quality. Here, we visually inspect those images generated at different training stages of a DCGAN model, on the MNIST and CIFAR-10 datasets.

Figures 11 and 12 show examples of generated CIFAR-10 and MNIST images respectively. As can be seen, the sample quality increases as training proceeds, and there is a strong correlation between decreasing CrossLID score and increasing sample quality. Note that the last subfigures in both Figure 11 and Figure 12 show the real images and the CrossLID score of the real data distribution (CrossLID($X_I; X_I$)), which is the lowest overall.

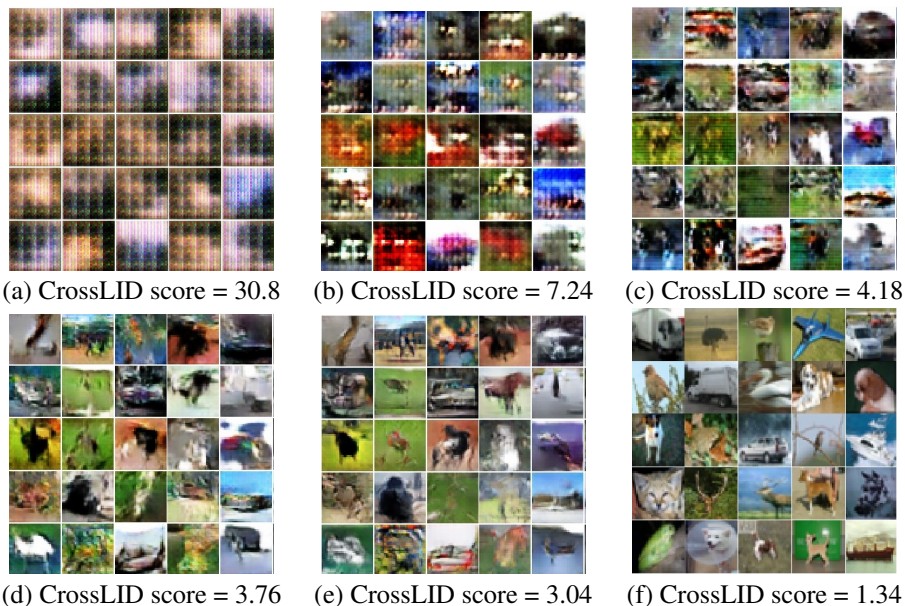

(a) CrossLID score = 30.8     (b) CrossLID score = 7.24     (c) CrossLID score = 4.18

(d) CrossLID score = 3.76     (e) CrossLID score = 3.04     (f) CrossLID score = 1.34

Figure 11: (a-e) 25 randomly selected DCGAN generated CIFAR-10 images and the CrossLID score of the DCGAN model after epoch 1, 5, 10, 20, and 49; (f) Real CIFAR-10 images and the CrossLID($X_I; X_I$) score.

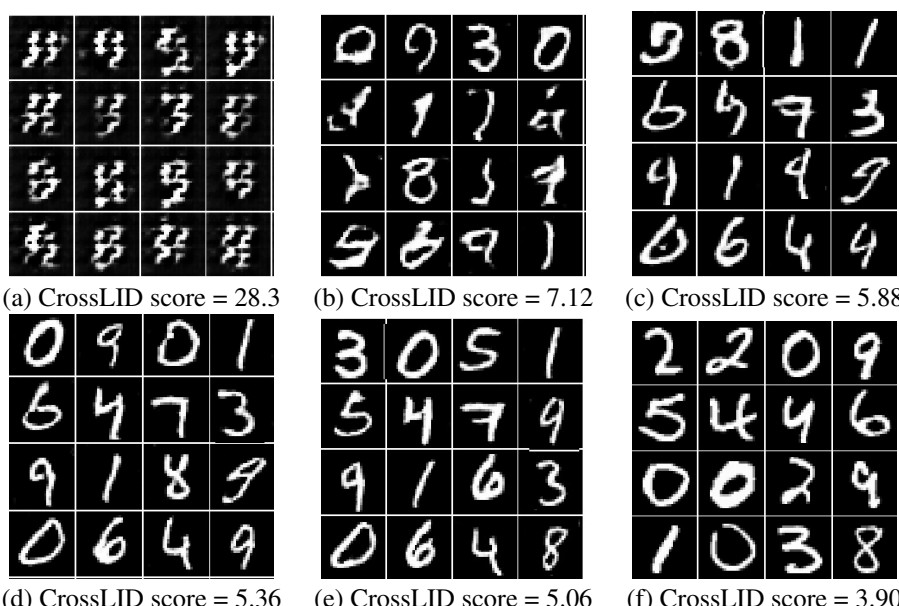

(a) CrossLID score = 28.3     (b) CrossLID score = 7.12     (c) CrossLID score = 5.88

(d) CrossLID score = 5.36     (e) CrossLID score = 5.06     (f) CrossLID score = 3.90

Figure 12: (a-e) 16 randomly selected DCGAN generated MNIST images and the CrossLID score of the DCGAN model after epoch 1, 4, 10, 20, and 50; (f) Real MNIST images and the CrossLID($X_I; X_I$) score.

## F    CORRELATION BETWEEN CROSSLID AND FID

In this section, we evaluate the correlation between the CrossLID and FID metrics. We calculate both metrics over different models of the MNIST, CIFAR10, and SVHN datasets. The models were taken from different epochs of a DCGAN training on the datasets. For each dataset, we compute two correlations between the FID and CrossLID scores (of different epochs): Pearson's correlation coefficient and Spearman's rank correlation coefficient. Table 3 shows the computed value of the coefficients. We observe a strong (but not perfect) correlation between the two metrics across all three datasets in terms of both correlation measures. Figure 13 shows the scatter plots of the scores for CIFAR10 dataset. Based on these results, we can say that the two measures show considerable level of agreement, but differences in the assessment of quality are nevertheless present, particularly for the more complex dataset CIFAR10 (correlations of roughly 0.8).

Table 3: Correlation coefficient between FID and CrossLID scores over different models across the datasets.

|  | MNIST | CIFAR10 | SVHN |
|---|---|---|---|
| Pearson | 0.99 | 0.79 | 0.69 |
| Spearman rank | 0.98 | 0.81 | 0.97 |

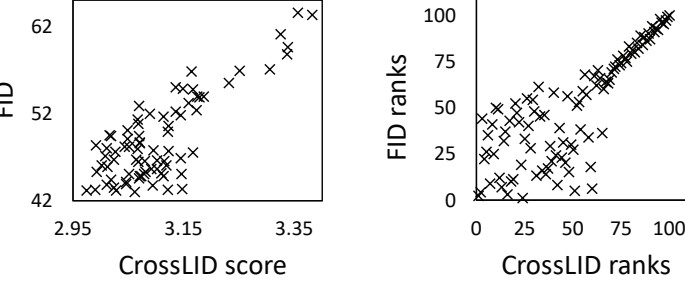

Figure 13: *Left*: Scatter plot of FID and CrossLID scores of different models over the CIFAR10 dataset. *Right*: Scatter plot of FID and CrossLID ranks of different models over the CIFAR10 dataset.

## G    NORMALIZED SCORES ON INPUT GAUSSIAN NOISE

For a better relative comparison of CrossLID, IS, and FID, we report here normalized values of all scores with respect to different levels of Gaussian noise applied to images. The scores are normalized to $[0, 1]$ by $\frac{S - S_0}{S_{100} - S_0}$ where $S$ is the score at a specific noise level, $S_0$ is the score at no noise, and $S_{100}$ is the score at maximum (100%) noise. Note that for all three metrics, a lower normalized score indicates better performance. The left subfigure of Figure 14 shows representative CIFAR10 images at various noise levels, confirming that the visual quality of the images is affected only slightly through the addition of noise. The right subfigure of Figure 14 shows the normalized scores for different levels of Gaussian noise (up to 2%). We observe that CrossLID is less sensitive than the other scores to low levels of input noise that do not greatly alter the visual quality of images. We can see that both IS and FID have high sensitivity to low level noise. Similar results for MNIST and SVHN datasets can be found in Figure 15. In our opinion, a high sensitivity to this low level of noise is undesirable, since the noisy images are highly recognizable. We note however that such a conclusion is necessarily subjective, as the GAN research community has not yet come to a consensus as to whether it is desirable for a quality measure to have high sensitivity to a level of noise which does not substantially alter the visual quality of an image. As a further comparison, in section H we study the behavior of GAN quality measures in the presence of very high levels of input noise.

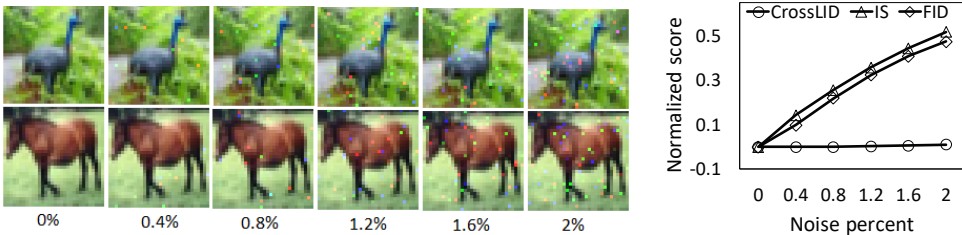

Figure 14: *Left*: Some representative CIFAR10 images after the application of different percentage of Gaussian noise. *Right*: Normalized CrossLID, IS, and FID under different levels of Gaussian noise.

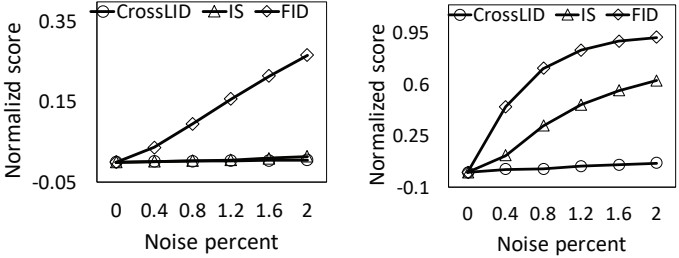

Figure 15: Normalized CrossLID, IS, and FID scores under different levels of Gaussian noise on MNIST (*Left*) and SVHN (*Right*) datasets.

## H FURTHER ROBUSTNESS EVALUATION ON INPUT NOISE

In addition to the evaluation of robustness against Gaussian input noise in Figure 5 of Section 6.1, we provide some analysis on one form of real-world image noise, the so-called 'salt-and-pepper' (or impulse) noise. The left subfigure of Figure 16 shows some representative images for low level salt-and-pepper noise, and the right subfigure shows the corresponding normalized scores of the three metrics CrossLID, IS, and FID. As shown in the left subfigure of Figure 16, the images are only slightly affected by the low levels of noise applied in the tests. For these low levels of noise, both the FID and IS scores increase considerably, whereas the CrossLID score remains stable.

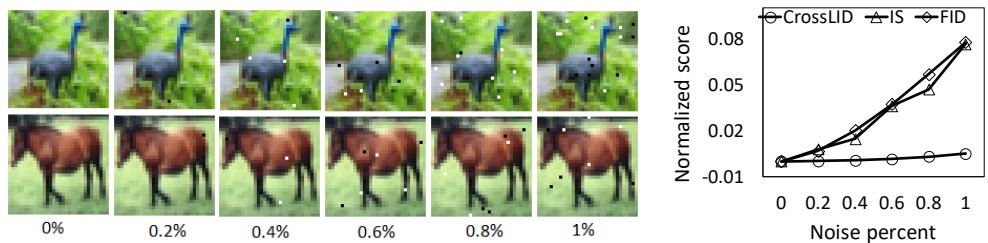

Figure 16: *Left*: representative CIFAR10 images after the application of different percentages of salt-and-pepper noise. *Right*: Normalized CrossLID, IS, and FID scores under low levels of salt-and-pepper noise.

A potential drawback of high sensitivity to low noise levels is that the metric may respond inconsistently for images with low noise as compared to images of extremely low quality. Consider Figure 17, wherein we report the three metrics for two specific types of noise: 2% Gaussian noise, and a black rectangle obscuring the center of the images. Although the images with Gaussian noise are visually superior to the other ones (with implanted rectangle), by virtue of its lower score, FID rates the obscured images to be of better quality — quite the opposite to human visual judgment. In contrast, for this particular scenario, the response of both CrossLID (for which a lower score indicates better quality) and Inception score (for which a higher score is better) is in line with human assessment of the visual quality of the images.

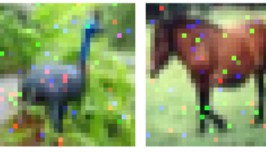 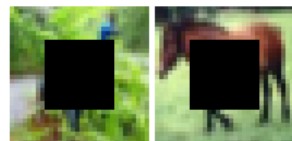

FID = 148.30
Inception score = 6.19
CrossLID score = 1.97

FID = 110.01
Inception score = 3.88
CrossLID score = 12.93

Figure 17: The CrossLID, Inception and FID scores of images with 2% Gaussian noise (*left*) and images with centers occluded by black rectangles (*right*).

We further show, in Figure 18, how the three metrics respond to the entire range of salt-and-pepper noise from 0% to 100%. Although all three scores increase as the noise level increases to 100%, they react in quite different ways. Both IS and FID are highly sensitive to low noise levels, while CrossLID is less sensitive. CrossLID increases sharply as noise begins to dominate (as seen in our results for 25% and 50% noise); from the representative images on the left, we see that this effect coincides with a drastic drop in visual quality. We believe that the response of CrossLID to different noise levels is more reasonably correlated with the visual quality of images, but note that this evaluation is necessarily subjective.

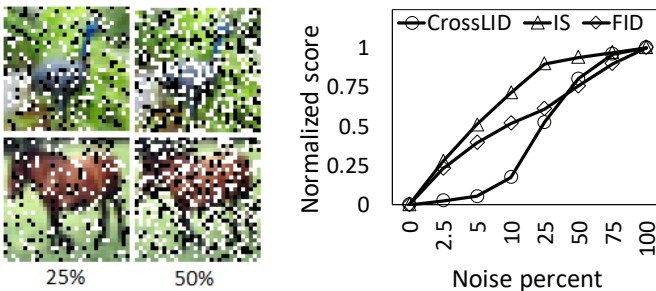

Figure 18: *Left*: Some representative CIFAR10 images with 25% and 50% salt-and-pepper noise. *Right*: Normalized CrossLID, FID, and Inception scores for images with high levels of salt-and-pepper noise. Note the non-linear scale on the $x$-axis.

## I    COMPARISON OF RUNNING TIME

We compare the running time of CrossLID, IS, and FID with respect to different sample sizes (from 10K to 50K) used for estimation. The running time was calculated as the time elapsed to obtain a score for a given set of samples, with an NVIDIA Titan V GPU. The left subfigure of Figure 19 shows the result on the CIFAR-10 dataset. As the sample size increases, the running time of the three metrics all increase in a linear fashion, but at different pace and scales. Among them, the CrossLID metric has the lowest running time while FID requires the highest. Note that the different feature extractors used by the three metrics may influence their running time slightly.

## J    IMPACT OF NEIGHBORHOOD SIZE ON CROSSLID ESTIMATION

The impact of neighborhood size $k$ on CrossLID estimation is illustrated in the right subfigure of Figure 19. For different choices of $k$, the estimated CrossLID scores all decrease as training progresses, which indicates that the estimates all accurately captured the improvement of GAN models over training. We also observe that a larger $k$ tends to result in a higher value of the estimate, an effect of the expansion of locality. Overall, the discriminability of CrossLID is not sensitive to different values of $k$.

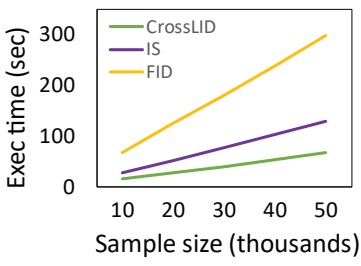 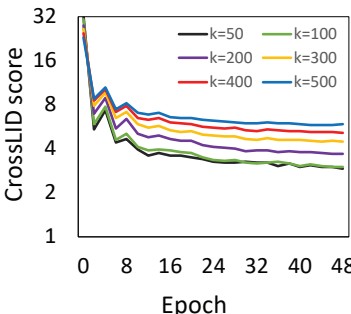

Figure 19: *Left*: Running time of CrossLID, Inception score (IS) and FID over different sample sizes on the CIFAR-10 dataset. *Right*: CrossLID scores estimated using different neighborhood size $k$, for images generated at different epochs by a DCGAN on the CIFAR-10 dataset.

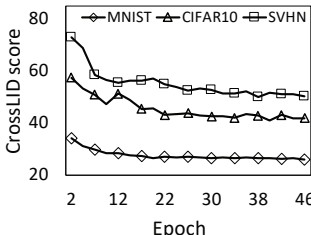

Figure 20: CrossLID scores estimated using discriminator features, over different epochs of DC-GAN training on MNIST, SVHN and CIFAR-10 datasets. Features of the third convolution layer of the discriminator network were used to compute the average CrossLID score of 20k generated images.

## K  CROSSLID ESTIMATION WITH GAN DISCRIMINATOR FEATURES

In this section, we evaluate the potential of a GAN discriminator as a feature extractor for CrossLID. For each of the MNIST, SVHN and CIFAR-10 datasets, we trained a DCGAN model for 50 epochs (architectures can be found in Appendix N.2), and use the discriminator learned at the end of training as the feature extractor for CrossLID. We generated 20K samples from the generators of each training epoch, and estimated the CrossLID score using the outputs of the third convolution layer of the discriminator as feature vectors. The results, shown in Figure 20, reveal that CrossLID can be estimated using GAN discriminator features without recourse to any external CNN classifier. As shown in Figure 20, the discriminator-estimated CrossLID scores followed the same trends as those scores estimated by external classifiers (see left subfigure in Figure 2, section 6.1). This at least partially confirms that CrossLID can be reliably estimated directly with the discriminator; a more detailed analysis is an interesting topic for future research.

## L  EVALUATION OF CROSSLID-GUIDED OVERSAMPLING APPROACH USING INCEPTION SCORE AND FID

Table 4 and 5 report the results in terms of Inception scores and FIDs, of the CrossLID guided oversampling experiments described in Section 6.2. We see that CrossLID guided oversampling methods (DCGAN+ and WGAN+) achieve lower FID results than those of standard GAN training (DCGAN and WGAN). Comparing the FID results with the CrossLID scores reported in Table 2, we find that their rankings are the same: the proposed oversampling approach achieved better results in terms of both metrics. This is the case with Inception scores as well, except for the SVHN dataset, for which both methods demonstrate similar results.

Table 4: Inception scores (mean±std over 5 random runs) of CrossLID guided oversampling.

| Dataset | Inception score (higher is better) | | | |
|---|---|---|---|---|
| | DCGAN | DCGAN+ | WGAN | WGAN+ |
| MNIST | $8.65 \pm 0.03$ | $\mathbf{8.76 \pm 0.01}$ | $8.13 \pm 0.04$ | $\mathbf{8.46 \pm 0.03}$ |
| CIFAR10 | $6.14 \pm 0.09$ | $\mathbf{6.32 \pm 0.09}$ | $5.46 \pm 0.03$ | $\mathbf{5.70 \pm 0.06}$ |
| SVHN | $\mathbf{3.03 \pm 0.03}$ | $3.01 \pm 0.02$ | $\mathbf{2.90 \pm 0.02}$ | $2.89 \pm 0.01$ |

Table 5: FID scores (mean±std over 5 random runs) of CrossLID guided oversampling.

| Dataset | FID (lower is better) | | | |
|---|---|---|---|---|
| | DCGAN | DCGAN+ | WGAN | WGAN+ |
| MNIST | $7.13 \pm 0.02$ | $\mathbf{6.49 \pm 0.04}$ | $16.40 \pm 0.06$ | $\mathbf{12.44 \pm 0.17}$ |
| CIFAR10 | $40.98 \pm 0.24$ | $\mathbf{39.75 \pm 0.12}$ | $56.06 \pm 0.24$ | $\mathbf{54.26 \pm 0.22}$ |
| SVHN | $13.87 \pm 0.07$ | $\mathbf{12.92 \pm 0.07}$ | $35.86 \pm 0.12$ | $\mathbf{32.82 \pm 0.20}$ |

## M  STABILITY OF THE PROPOSED GAN TRAINING WITH OVERSAMPLING

Here, we provide more details regarding the training of DCGANs without batch normalization layers 1) in the generator, or 2) in both the generator and the discriminator. Following the methodology used in (Arjovsky et al., 2017) to verify model stability, we illustrate how our proposed oversampling strategy can avoid mode collapse and help learning, by showing for the MNIST dataset the images generated at different epochs. As visualized in Figure 21, when batch normalization layers were removed from both the generator and the discriminator, standard training suffered from mode collapse from the beginning of training, and the images generated images by the end of training were of low quality. More severe mode collapse was observed with standard training when the batch normalization layers were removed from the discriminator only: in this case, the model failed to generate realistic images (see Figure 22). On the other hand, when trained with our proposed oversampling strategy, mode collapse was not observed in any of the two scenarios, and the generated images were of higher quality consistently throughout training.

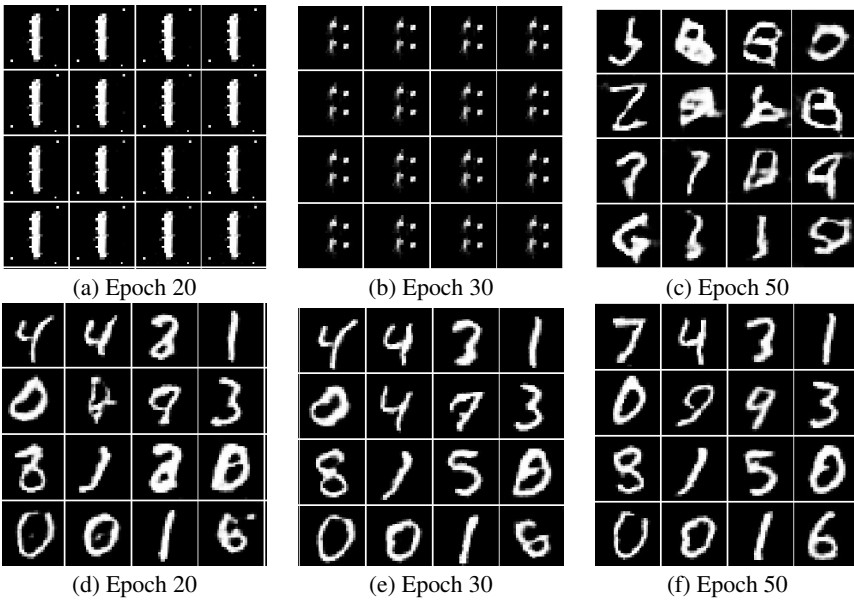

(a) Epoch 20     (b) Epoch 30     (c) Epoch 50

(d) Epoch 20     (e) Epoch 30     (f) Epoch 50

Figure 21: MNIST images generated at epoch 20, 30 and 50 (50 epochs in total) by DCGAN models without batch normalization layers in both the generator and discriminator. *Top row*: Images generated by a DCGAN model trained using standard training. *Bottom row*: Images generated by a DCGAN model trained with our proposed oversampling strategy.

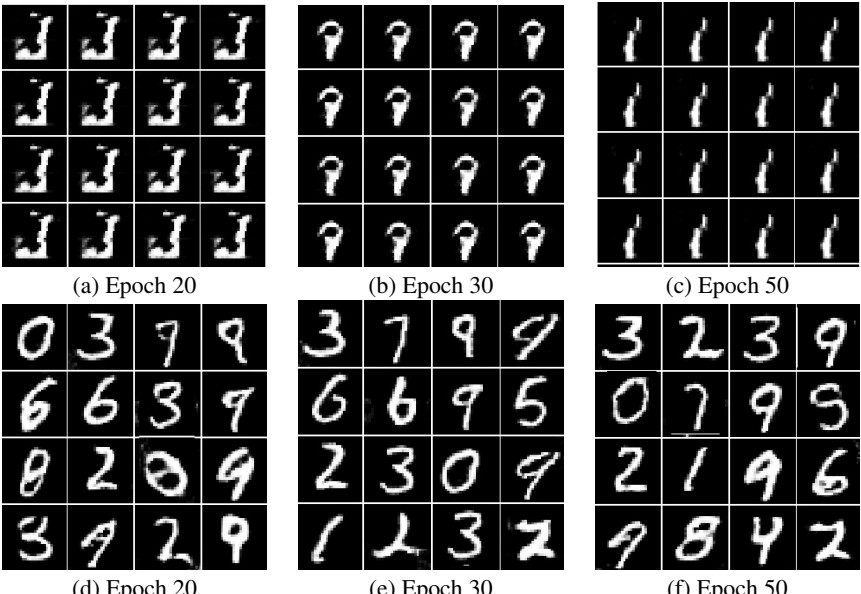

(a) Epoch 20          (b) Epoch 30          (c) Epoch 50

(d) Epoch 20          (e) Epoch 30          (f) Epoch 50

Figure 22: MNIST images generated at epoch 20, 30 and 50 (50 epochs in total) by DCGAN models without batch normalization layers in the discriminator (the generator network still has batch normalization). *Top row*: Images generated by a DCGAN model trained using standard training. *Bottom row*: Images generated by a DCGAN model trained with our proposed oversampling strategy.

# N    NETWORK ARCHITECTURES AND EXPERIMENTAL SETTINGS

## N.1    GAN ARCHITECTURE, TRAINING DETAILS AND EXEMPLARY OUTPUTS

On each dataset, we used the same architecture for both DCGAN and WGAN, following the architectural guidelines in Radford et al. (2016). The generator and discriminator networks used for MNIST and CIFAR-10/SVHN are described in Table 6 and Table 7 respectively. Note that the Tanh output activation was removed from the WGAN generator so as to produce a linear output for the training of WGANs with the Wasserstein loss.

The DCGANs were trained for 60 and 100 epochs on MNIST and CIFAR-10/SVHN respectively, using the Adam optimizer (Kingma & Ba, 2015) with learning rate 0.0001 and decay 0.00001. The WGANs models were trained for 200 epochs on all datasets using the RMSProp optimizer. A learning rate of 0.00005 was used for CIFAR10 and SVHN training, while 0.0001 was used for MNIST. For our proposed training strategy, we used $N_1 = 20K$, $N_2 = 2K$ for class-wise CrossLID estimation, and sample size $m = 30K$ for DCGANs and $m = 20K$ for WGANs. For $T$, we simply used the number of generator iterations in one epoch, i.e., we applied oversampling after every epoch. Figure 23 shows some randomly selected images generated by 1) DCGANs trained using standard training versus 2) DCGANs trained with our proposed oversampling strategy.

## N.2    EXTERNAL CNN MODELS USED FOR FEATURE EXTRACTION

Table 8 describes the architectures of external CNN models used for feature extraction on MNIST, CIFAR-10 and SVHN datasets; the selected feature layers are highlighted in **bold**. These CNN classifiers were trained separately on the original training sets of the three datasets. For MNIST and SVHN networks, the outputs of the first fully connected (FC) layer was used as features, while for CIFAR-10, the output of the last max pooling layer was used. For the estimation of our proposed CrossLID, the networks were applied to extract the features for both real and fake images, and the extracted features were then used to compute the CrossLID scores.

Table 6: The generator and discriminator network used for the MNIST dataset. Conv($x, y, z$) represents a convolution layer with $x$ filters of kernel size $y \times y$ and stride $z$. ConvTr($x, y, z$) represents a transposed convolution layer with $x$ filters of kernel size $y \times y$ and stride $z$. FC($x$) represents a fully connected layer with $x$ output nodes. BN represents a batch normalization layer, R represents the reshape operation and LReLU is the LeakyRelu.

| Generator | Discriminator |
|---|---|
| Input: Z(100) | Input: (28,28,1) |
| R(1,1,100) | Conv(32,3,2), BN, LReLU |
| ConvTr(128,3,1), BN, ReLU | Conv(64,3,2), BN, LReLU |
| ConvTr(64,3,2), BN, ReLU | Conv(128,3,2), BN, LReLU |
| ConvTr(32,3,2), BN, ReLU | Conv(1,3,1), Sigmoid |
| ConvTr(1,3,2), Tanh | Output: 1 |
| Output: (28, 28, 1) | |

Table 7: The generator and discriminator network used for the CIFAR-10 and SVHN datasets. Conv($x, y, z$) represents a convolution layer with $x$ filters of kernel size $y \times y$ and stride $z$. ConvTr($x, y, z$) represents a transposed convolution layer with $x$ filters of kernel size $y \times y$ and stride $z$. FC($x$) represents a fully connected layer with $x$ output nodes. BN represents a batch normalization layer, R represents the reshape operation and LReLU is the LeakyRelu

| Generator | Discriminator |
|---|---|
| Input: Z(100) | Input: (32,32,3) |
| R(1,1,100) | Conv(64,4,2), BN, LReLU |
| ConvTr(256,4,1), BN, ReLU | Conv(128,4,2), BN, LReLU |
| ConvTr(128,4,2), BN, ReLU | Conv(256,4,2), BN, LReLU |
| ConvTr(64,4,2), BN, ReLU | Conv(1,4,1), Sigmoid |
| ConvTr(3,4,3), Tanh | Output: 1 |
| Output: (32, 32, 3) | |

Table 8: Network architecture of external CNN models used for feature extraction. Conv($x, y, z$) represents a convolution layer with $x$ filters of kernel size $y \times y$ and stride=$z$. MaxPool($x, y$) represents a max-pooling layer with pool size $x \times y$. FC($x$) represents a fully connected layer with $x$ output nodes. The selected feature layers are highlighted in **bold**.

| Dataset | Architecture of external CNN models |
|---|---|
| MNIST | Conv(32,3,1), Conv(64,3,1), MaxPool(2,2), **FC(128)**, FC(10) |
| SVHN | Conv(32,3,1), Conv(32,3,1), MaxPool(2,2), Conv(64,3,1), Conv(64,3,1), MaxPool(2,2), Conv(128,3,1), Conv(128,3,1), Max-Pool(2,2), **FC(512)**, FC(10) |
| CIFAR-10 | Conv(64,3,1), Conv(64,3,1), MaxPool(2,2), Conv(128,3,1), Conv(128,3,1), MaxPool(2,2) Conv(256,3,1), Conv(256,3,1), Conv(256,3,1), MaxPool(2,2), Conv(512,3,1), Conv(512,3,1), Conv(512,3,1), MaxPool(2,2), Conv(512,3,1), Conv(512,3,1), Conv(512,3,1), **MaxPool(2,2)**, FC(512), FC(10) |

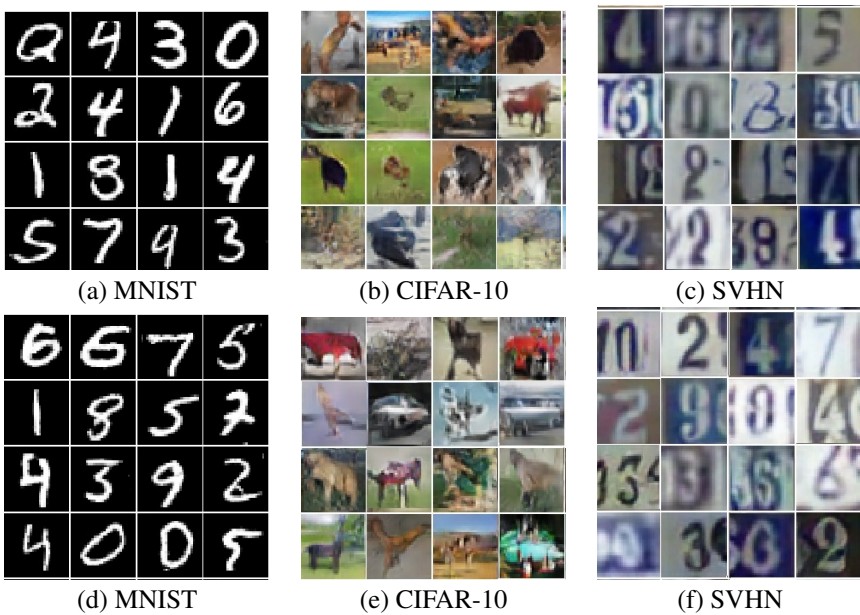

(a) MNIST     (b) CIFAR-10     (c) SVHN

(d) MNIST     (e) CIFAR-10     (f) SVHN

Figure 23: *Top row:* Images generated by standard DCGAN training (without our proposed oversampling) on the three datasets (a-c). *Bottom row:* Images generated by DCGANs trained with our proposed oversampling strategy on the three datasets (d-f).

