# OpenReview forum: "Quality Evaluation of GANs Using Cross Local Intrinsic Dimensionality"
_ICLR.cc/2019/Conference_

### Official Review · AnonReviewer3 · 2018-10-28
**Sample-based quantitative evaluation of generative models based on k-nearest neighbor queries from the observed samples into the generated samples in a learned feature space.**

**Rating:** 6
**Confidence:** 5

**Review:**

Statistics based on KNN distances are ubiquitous in machine learning. In this paper the authors propose to apply the existing LID metric to GANs. The metric can be decomposed as follows: (1) Given a point x in X, compute the k-nearest neighbors KNN(x, X) and let those distances be R1, R2, …, Rk. Now, rewrite LID(x, X) = [max_over_i (log Ri) - mean_over_i (log Ri)] to uncover that the distribution of (log-)distances is summarized as a function of the max distance and the mean distance. (2) To extend the metric to two sets, A and B, define CrossLID(A; B) = E_(x in A) [LID(x, B)]. To see why CrossLID is useful, let X be the observed data and G the generated data. First consider CrossLID(A, B) where A=B=X which determines a lower-bound which is essentially the average (over elements of A) LID statistic determined by the underlying KNN graph of X. Now, keep A=X, and progressively change B to G (say by replacing some points from X with some points from G). This will induce a change of the distance statistics of some points from A, which will be detected on the individual LID scores of those points, and will hence be propagated to CrossLID. As a result, LID close to the baseline LID detects both sample quality issues as well as mode dropping/collapse issues. In practice, instead of computing this measure in the pixel space, one can compute it in the feature space of some feature extractor, or in some cases directly in the learned feature space of the generator. Finally, given some labeling of the points, one can keep track of the CrossLID statistic for each mode and use this during training to oversample modes for which the gap between the expected CrossLID and computed one is large.

Clarity: I think the clarity can be improved -- instead of stating the (rather abstract) properties of LID, the readers might benefit from the direct discussion of the LID estimator and a couple of examples, derive the max - mean relationship for the MLE estimator and provide some guiding comments. In a later section one might discuss why the estimator is so powerful and generally applicable. Secondly, the story starts with “discriminability of the distance measure” and the number of latent variables needed to do it, but I felt that this only complicated matters as many of these concepts are unclear at this point.
Originality: Up to my knowledge, the proposed application is novel, albeit built on an existing (well-known) estimator. Nevertheless, the authors have demonstrated several desirable properties which might be proven useful in practice.
Significance of this work: The work is timely and attempts to address a critical research problem which hinders future research on deep generative models.

Pro:
- Generally well written paper, although the clarity of exposition can be improved.
- Estimator is relatively easy to compute in practice (i.e. the bottleneck will still be in the forward and backward passes of the DNNs).
- Can be exploited further when labeled data is available
- Builds upon a strong line of research in KNN based estimators.
- Solid experimental setup with many ablation studies.

Con:
- FID vs CrossLID: I feel that many arguments against FID are too strong. In particular, in “robustness to small input noise” and “robustness to input transformation” you are changing the underlying distribution *significantly* -- why should the score be invariant to this? After all, it does try to capture the shift in distribution. In the robustness to sample size again FID is criticized to have a high-variance in low-sample size regime: This is well known, and that’s why virtually all work presenting FID results apply 10k samples and average out the results over random samples. In this regime it was observed that it has a high bias and low variance (Figure 1 in [1]). In terms of the dependency of the scores to an external model, why wouldn’t one compute FID on the discriminator feature space? Similarly, why wouldn’t one compute FID in the pixel space and get an (equally bad) score as LID in pixel space? Given these issues, in my opinion, Table 1 overstates the concerns with FID, and understates the issues with CrossLID.
- FID vs CrossLID in practice: I argue that the usefulness comes from the fact that relative model comparison is sound. From this perspective it is critical to show that the Spearman’s rank correlation between these two competing approaches on real data sets is not very high -- hence, there are either sample quality or mode dropping/collapsing issues detected by one vs the other. Now, Figure 1 in [1] shows that this FID is sensitive to mode dropping. Furthermore, FID is also highly correlated with sample quality (page 7 of [2]).
- A critical aspect here is that in pixel space of large dimension the distances will tend to be very similar, and hence all estimators will be practically useless. As such, learning the proper features space is of paramount importance. In this work the authors suggest two remedies: (1) Compute a feature extractor by solving a surrogate task and have one extractor per data set. (2) During the training of the GAN, the discriminator is “learning” a good feature space in which the samples can be discriminated. Both of these have significant drawbacks. For (1) we need to share a dataset-specific model with the community. This is likely to depend on the preprocessing, model capacity, training issues, etc.. Then, the community has to agree to use one of these. On the other hand, (2) is only useful for biasing a specific training run. Hence, this critical aspect is not addressed and the proposed solution, while sensible, is unlikely to be adopted.
- Main contributions section is too strong -- avoiding mode collapse was not demonstrated. Arguably, given labeled data, the issue can be somewhat reduced if the modes correspond to labels. Similarly, if the data is well-clusterable one can expect a reduction of this effect. However, as both the underlying metric as well as the clustering depends on the feature space, I believe the claim to be too strong. Finally, if we indeed have labels or some assumptions on the data distribution, competing approaches might exploit it as well (as done with i.e. conditional GANs).
- In nonparametric KNN based density estimation, one often uses statistics based on KNN distances. What is the relation to LID?

With respect to the negative points above, without having a clear cut case why this measure outperforms and should replace FID in practice, I cannot recommend acceptance as introducing yet another measure might slow down the progress. To make a stronger case I suggest:
(1) Compute Spearman's rank correlation between FIDs and CrossLIDs of several trained models across these data sets.
(2) Compute the Pearson's correlation coefficient across the data sets. Given that your method has access to dataset specific feature extractors I expect it perform significantly better than FID.

[1] https://arxiv.org/pdf/1711.10337.pdf
[2] https://arxiv.org/pdf/1806.00035.pdf

========
Thank you for the detailed responses. I have updated my score from 5 to 6.

---

> ### Author Response · Authors · 2018-11-15
> **Response to Reviewer 3 - Part 3**
>
> --" Main contributions section is too strong -- avoiding mode collapse was not demonstrated. Arguably, given labeled data, the issue can be somewhat reduced if the modes correspond to labels. Similarly, if the data is well-clusterable one can expect a reduction of this effect. However, as both the underlying metric as well as the clustering depends on the feature space, I believe the claim to be too strong. Finally, if we indeed have labels or some assumptions on the data distribution, competing approaches might exploit it as well (as done with i.e. conditional GANs).”
>
> RESPONSE 3.8: Mode collapse avoidance was evidenced in the paper in Section 6.2 (“Effectiveness in Preventing Mode Collapse”) and in more detail in Appendix M.
> We have reworded the main contributions section.  Please see comment G2 above.
> We do not believe it is possible to deploy an oversampling procedure with FID on classic GAN models.   The reason is that for a given class, FID would need to compare the GAN distribution samples with the real distribution samples.  However, there is no class label available for the GAN distribution samples, and so such comparison isn’t possible.     In contrast, CrossLID only requires availability of class labels for the real distribution samples and hence the oversampling approach can be used.
> Estimation of CrossLID is possible mode-wise, because the estimation is essentially “local”.  One just computes the nearest neighbours of a real sample, using neighbours from the entire GAN distribution.  Then an average is taken over all real samples within a class. In contrast, estimation of FID is essentially “global”, and operates by comparing two distributions. Mode-wise computation of FID may be possible for specific types of GAN models such as conditional GANs. The benefit of CrossLID is that it does not require any such specific GANs for mode-wise performance estimation.  In regard to conditional GANs,  this is a good suggestion for future work and we have added a note about this in future work of the paper.
>
>
> --" In nonparametric KNN based density estimation, one often uses statistics based on KNN distances. What is the relation to LID?”
>
> RESPONSE 3.9: KNN-based density estimation makes explicit use of  the volume of the m-dimensional ball with radius equal to that of the neighborhood. The LID model, on the other hand, is concerned with the order of magnitude (or scale) of the growth rate in probability measure (not volume) as neighborhoods are expanded. As such, LID has the very useful property of being oblivious to the representational dimension of the data domain. We have updated Section 3 of the paper so as to emphasize the lack of dependence on knowledge of the representational dimension.
>
>
> --"With respect to the negative points above, without having a clear cut case why this measure outperforms and should replace FID in practice, I cannot recommend acceptance as introducing yet another measure might slow down the progress. To make a stronger case I suggest:
> (1) Compute Spearman's rank correlation between FIDs and CrossLIDs of several trained models across these data sets.
> (2) Compute the Pearson's correlation coefficient across the data sets. Given that your method has access to dataset specific feature extractors I expect it perform significantly better than FID.”
>
> RESPONSE: 3.10:  We have included these results in Appendix F of the paper.  We found that CrossLID and FID scores are well-correlated (but not perfectly correlated) for all datasets in terms of both Pearson’s coefficients and Spearman’s rank coefficients. Although, the two metrics are well correlated, simple scatter plots of the scores and ranks show that there are subtle differences in their exact rankings of the models and thus each measure is providing a different perspective about the GAN generated data.   We believe this validates the introduction of CrossLID - it is well correlated with FID, but not exactly and thus provides a further additional perspective about GAN sample quality.

---

> ### Author Response · Authors · 2018-11-15
> **Response to Reviewer 3 - Part 2**
>
> --"In terms of the dependency of the scores to an external model, why wouldn’t one compute FID on the discriminator feature space? Similarly, why wouldn’t one compute FID in the pixel space and get an (equally bad) score as LID in pixel space? Given these issues, in my opinion, Table 1 overstates the concerns with FID, and understates the issues with CrossLID.”
>
>
> RESPONSE 3.5: Please see our responses G2 above.  Although we believe exploring this issue is beyond the scope of our study, we have done some preliminary experiments on FID in a discriminator feature space, and the results indicate that FID can be computed as well using the discriminator activations.  We have updated the “Dependency on external model” in section 6.1 and Table 1 of the paper accordingly to reflect that FID might also be computed on the discriminator feature space.
>
>
> --" FID vs CrossLID in practice: I argue that the usefulness comes from the fact that relative model comparison is sound. From this perspective it is critical to show that the Spearman’s rank correlation between these two competing approaches on real data sets is not very high -- hence, there are either sample quality or mode dropping/collapsing issues detected by one vs the other. Now, Figure 1 in [1] shows that this FID is sensitive to mode dropping. Furthermore, FID is also highly correlated with sample quality (page 7 of [2]).”
>
> RESPONSE 3.6 For correlation tests, please see our response 3.10 below.   We agree that it has been shown in the literature that FID is sensitive to mode dropping and correlated with sample quality.   Indeed FID also performs well on these scenarios in our paper (See “Sensitivity to mode collapse” section and Table 1).    Please also see  our response G1 above - we are *not* arguing that one needs to “replace” FID.
>
> --" A critical aspect here is that in pixel space of large dimension the distances will tend to be very similar, and hence all estimators will be practically useless. As such, learning the proper features space is of paramount importance. In this work the authors suggest two remedies: (1) Compute a feature extractor by solving a surrogate task and have one extractor per data set. (2) During the training of the GAN, the discriminator is “learning” a good feature space in which the samples can be discriminated. Both of these have significant drawbacks. For (1) we need to share a dataset-specific model with the community. This is likely to depend on the preprocessing, model capacity, training issues, etc.. Then, the community has to agree to use one of these. On the other hand, (2) is only useful for biasing a specific training run. Hence, this critical aspect is not addressed and the proposed solution, while sensible, is unlikely to be adopted.”
>
> RESPONSE 3.7: We have updated the paper to note debate around these issues.    Our view is this - the community could choose to use CrossLID in different ways - it could use a dataset specific feature extractor (as we have outlined), or it might choose to use a single feature extractor based on the Inception model (like what is done for FID and LID).  We preferred to evaluate the proposed measure on dataset-specific feature extractors as they give more robust feature vectors, and the evaluation metric is expected to have high discriminability. The drawback of using a single feature extractor, e.g., Inception, is that for a very different data distribution, e.g., SVHN, the features may not be robust enough. Authors in (Shane Barratt and Rishi Sharma, 2018) also provided similar arguments in favor of domain-specific feature extractors. Regarding (2), we agree that training specific bias may be induced. Indeed, our intention was to only show that the discriminator feature space may work in practice for feature extraction when no external model is available. Thus, in case of unavailability of an external feature extractor, for example, in case of a non-image, unlabeled dataset, one may find the approach useful for computing the value of an evaluation metric.

---

> ### Author Response · Authors · 2018-11-15
> **Response to Reviewer 3 - Part 1**
>
> Thank you very much for your comments. Please read the above “General Response to Reviewers 1, 2 and 3” for our general response G1, G2 and G3.
>
> --"Clarity: I think the clarity can be improved -- instead of stating the (rather abstract) properties of LID, the readers might benefit from the direct discussion of the LID estimator and a couple of examples, derive the max - mean relationship for the MLE estimator and provide some guiding comments. In a later section one might discuss why the estimator is so powerful and generally applicable. Secondly, the story starts with “discriminability of the distance measure” and the number of latent variables needed to do it, but I felt that this only complicated matters as many of these concepts are unclear at this point.”
>
> RESPONSE 3.1:  The reviewer is correct in regarding the MLE estimator of LID as determined by (the reciprocal of) the difference between the maximum and mean of the log-distances within a neighborhood sample. This difference is basically how the estimator assesses the form of discriminability modeled by LID. While we still feel that it is important to introduce LID from a correct, theoretical perspective (from its motivation in terms of intrinsic dimensionality & discriminability), we agree that this practical explanation of the MLE estimator is helpful too. We have added an explanation in terms of max and mean log-distances to Section 3 (in the context of LID, immediately after the statement of the MLE estimator), Section 4.1 (in the context of CrossLID, in the first paragraph after its definition), and Section 4.2 (in the statement of Equation 7).
>
>
> --" Estimator is relatively easy to compute in practice (i.e. the bottleneck will still be in the forward and backward passes of the DNNs).”
>
> RESPONSE 3.2:  In fact, only one forward pass is required to compute CrossLID measure.  No backward pass is needed for feature extraction.  We have added mention of this in Section 4.2 of the updated paper.
>
> --"Con:
> - FID vs CrossLID: I feel that many arguments against FID are too strong. In particular, in “robustness to small input noise” and “robustness to input transformation” you are changing the underlying distribution *significantly* -- why should the score be invariant to this? After all, it does try to capture the shift in distribution.”
>
> RESPONSE 3.3: Please see our responses G3 above. This is an interesting comment.   We believe that debate over properties of an ideal GAN quality measure is extremely important for the ICLR community.   We do not claim our perspective is the only one, but we do believe it adds an important new dimension to the discussion (and after all, such type of discussion should be one of the purposes of a paper at ICLR).  We have added some extra discussion to “Robustness to small input noise” section of the updated paper about this issue.   See also general comment G3 above.  A potential drawback of high sensitivity to low noise levels is that the metric may respond inconsistently for images with low noise as compared to images of extremely low quality. For which we show in Fig. 17, FID rates the images with centers occluded by black rectangles to be of better quality than images with 2% Gaussian noise --- quite the opposite to human visual judgment.
>
>
> --"In the robustness to sample size again FID is criticized to have a high-variance in low-sample size regime: This is well known, and that’s why virtually all work presenting FID results apply 10k samples and average out the results over random samples. In this regime it was observed that it has a high bias and low variance (Figure 1 in [1]).”
>
> RESPONSE 3.4: We have added a comment to the “Robustness to sample size” section in the experiments of the updated paper noting that this methodology is already being used in the literature ([1]) when deploying FID.
> Note that in all our experiments related to FID we used a sample size of 50k, which is large enough for FID to get a reasonably accurate estimation.

---

### Official Review · AnonReviewer2 · 2018-11-04
**Authors coupled a local intrinsic dimensionality measure to assess GAN frameworks concerning their ability to generate realistic data. The proposal is straightforward and would be applied in different GAN-based approaches, mainly, being sensitive to mode collapse.**

**Rating:** 6
**Confidence:** 4

**Review:**

The paper is clear regarding motivation, related work, and mathematical foundations. The introduced cross-local intrinsic dimensionality- (CLID) seems to be naive but practical for GAN assessment. In general, the experimental results seem to be convincing and illustrative.

Pros:
- Clear mathematical foundations and fair experimental results.
- CLID can be applied to favor GAN-based training, which is an up-to-date research topic.
- Robustness against mode collapse (typical discrimination issue).

Cons:
-The CLID highly depends on the predefined neighborhood size, which is not studied properly during the paper. Authors suggest some experimentally fixed values, but a proper analysis (at least empirically), would be useful for the readers.
- The robustness against input noise is studied only for small values, which is not completely realistic.

---

> ### Author Response · Authors · 2018-11-15
> **Response to Reviewer 2**
>
> Thank you very much for your comments.
>
> --"Cons:
> -The CLID highly depends on the predefined neighborhood size, which is not studied properly during the paper. Authors suggest some experimentally fixed values, but a proper analysis (at least empirically), would be useful for the readers.
> - The robustness against input noise is studied only for small values, which is not completely realistic.”
>
> RESPONSE 2.1:  We have included the results of “impact of neighborhood size on CrossLID estimation” in Appendix J and “impact of high level noise” in Appendix H (please see Fig. 18 and relevant discussion in Appendix H) of the updated paper.

---

### Official Review · AnonReviewer1 · 2018-11-06
**Concerns about clarity and scalability of the metric**

**Rating:** 4
**Confidence:** 3

**Review:**

The paper proposes a new metric to evaluate GANs. The metric, Cross Local Intrinsic Dimensionality (CrossLID) is estimated by comparing distributions of nearest neighbor distances between samples from the real data distribution and the generator. Concretely, it proposes using the inverse of the average of the negative log of the ratios of the distances of the K nearest neighbors to the maximum distance within the neighborhood.

The paper introduces LID as the metric to be used within the introduction, but for readers unfamiliar with it, the series of snippets “model of distance distributions” and “assesses the number of latent variables” and “discriminability of a distance measure in the vicinity of x”  are abstract and lack concrete connections/motivations for the problem (sample based comparison of two high-dimensional data distributions) the paper is addressing.

After an effective overview of relevant literature on GAN metrics, LID is briefly described and motivated in various ways. This is primarily a discussion of various high-level properties of the metric which for readers unfamiliar with the metric is difficult to concretely tie into the problem at hand. After this, the actual estimator of LID used from the literature (Amsaleg 2018) is introduced. Given that this estimator is the core of the paper, it seems a bit terse that the reader is left with primarily references to back up the use of this estimator and connect it to the abstract discussion of LID thus far.

Figure 1 is a good quick overview of some of the behaviors of the metric but it is not clear why the MLE estimator of LID should be preferred (or perform any differently) on this toy example from a simple average of 1-NN distances. The same is also appears to be true for the motivating example in Figure 8 as well.

To summarize a bit, I found that the paper did not do the best job motivating and connecting the proposed metric to the task at hand and describing in an accessible fashion its potentially desirable properties.

The experimental section performs a variety of comparisons between CrossLID, Inception Score and FID. The general finding of the broader literature that Inception Score has some undesirable properties is confirmed here as well. A potentially strong result showing where CrossLID performs well at inter-class mode dropping, Figure 4, is unfortunately confounded with sample size as it tests FID in a setting using 100x lower than the recommended amount of samples.

The analysis in this section is primarily in the form of interpretation of visual graphs of the behavior of the metrics as a quantity is changed over different datasets. I have some concerns that design decisions around these graphs (normalizing scales, subtracting baseline values) could substantially change conclusions.

An oversampling algorithm based on CrossLID is also introduced which results in small improvements over a baseline DCGAN/WGAN and improves stability of a DCGAN when normalization is removed. A very similar oversampling approach could be tried with FID but is not - potentially leaving out a result demonstrating the effectiveness of CrossLID.

The paper also proposes computing CrossLID in the feature space of a discriminator to make the metric less reliant on an external model. While this is an interesting thing to showcase - FID can also be computed in an arbitrary feature space and the authors do not clarify or investigate whether FID performs similarly.

These two extensions, addressing mode collapse via oversampling and using the feature space of a discriminator are interesting proposals in the paper, but the authors do not do a thorough investigation of how CrossLID performs to FID here.

Several experiments get into some unclear value judgements over what the behavior of an ideal metric should be. The authors of FID argue the opposite position of this paper that the metric should be sensitive to low-level changes in addition to high-level semantic content. It is unclear to me as the reader which side to take in this debate.

I have some final concerns over the fact that the metric is not tested on open problems that GANs still struggle with. Current SOTA GANs can already generate convincing high-fidelity samples on MNIST, SVHN, and CIFAR10. Exclusively testing a new metric for the future of GAN evaluation on the problems of the past does not sit well with me.

Some questions:
* Could the authors comment on run time comparisons of the metric with FID/IS?
* How much benefit is there from something like CrossLID compared to the simplest case of distance to 1-NN in feature space? More generally an analysis of how the benefits of CrossLID as you increase neighborhood size would help illuminate the behavior of the metric.
* For Table 2, what are the FID scores and how do they correlate with CrossLID and Inception Score?

Pros:
+ Code is available!
+ The metric appears to be more robust than FID in small sample size settings.
+ A variety of comparisons are made to several other metrics on three canonical datasets.
+ The paper has two additional contributions in addition to the metric. Addressing mode collapse via adaptive oversampling and utilizing the features of the discriminator to compute the metric in.
Cons:
- No error bars / confidence intervals are provided to show how sensitive the various metrics tested are to sample noise.
- Authors test FID outside of recommended situations (very low #of samples (500) in Figure 4) without noting this is the case. The stated purpose of Figure 4 is to evaluate inter-class mode dropping yet this result is confounded by the extremely low N (100x lower than the recommended N for FID).
- It is unclear whether metric continues to be reliable for more complex/varied image distributions such as Imagenet (see main text for more discussion)
- Many of the proposed benefits of the model (mode specific dropping and not requiring an external model) can also be performed for FID but the paper does not note this or provide comparisons.

---

> ### Author Response · Authors · 2018-11-15
> **Response to "Concerns about clarity and scalability of the metric" - Part 3**
>
> -- "* How much benefit is there from something like CrossLID compared to the simplest case of distance to 1-NN in feature space? More generally an analysis of how the benefits of CrossLID as you increase neighborhood size would help illuminate the behavior of the metric.”
>
> RESPONSE 1.9: We have not included the 1-NN distance in our study for the reasons laid out in Response 1.1. Also, in Section 4.1 of our updated version we now note that in Amsaleg et al. (2018), an estimator of local intrinsic dimensionality using only the 1-NN and k-NN distance measurements (a variant in the ‘MiND’ family) was shown to lead to relatively poor performance.
>
>
> -- "* For Table 2, what are the FID scores and how do they correlate with CrossLID and Inception Score?”
>
> RESPONSE 1.10: We have updated Table 2 of the paper and included FID and Inception results in Appendix L of the updated  paper. In addition, we have also added standard deviations for all metrics in Table 2 and Appendix L. The results indicate that,  for all datasets, FID and CrossLID scores rank the competitive methods of our experiments similarly. Inception score is also consistent with FID and CrossLID scores, except on the SVHN dataset. The discussions are included in Appendix L.
>
>
> -- "Cons:
> - No error bars / confidence intervals are provided to show how sensitive the various metrics tested are to sample noise.
>
> - Authors test FID outside of recommended situations (very low #of samples (500) in Figure 4) without noting this is the case. The stated purpose of Figure 4 is to evaluate inter-class mode dropping yet this result is confounded by the extremely low N (100x lower than the recommended N for FID).
> - It is unclear whether metric continues to be reliable for more complex/varied image distributions such as Imagenet (see main text for more discussion)
> - Many of the proposed benefits of the model (mode specific dropping and not requiring an external model) can also be performed for FID but the paper does not note this or provide comparisons.”
>
> RESPONSE 1.11:  Addressed above, please see 1.5, 1.8, G3. For all metrics, the standard deviation is very small. We have added standard deviations for all metrics in Table 2 and Appendix L of GAN experimental results.. We hope these results will help the reader to understand the confidence interval of the metrics in general.

---

> ### Author Response · Authors · 2018-11-15
> **Response to "Concerns about clarity and scalability of the metric" - Part 2**
>
> -- "An oversampling algorithm based on CrossLID is also introduced which results in small improvements over a baseline DCGAN/WGAN and improves stability of a DCGAN when normalization is removed. A very similar oversampling approach could be tried with FID but is not - potentially leaving out a result demonstrating the effectiveness of CrossLID.”
>
> RESPONSE 1.4: We do not believe it is possible to deploy an oversampling procedure with FID on classic GAN models.   The reason is that for a given class, FID would need to compare the GAN distribution samples with the real distribution samples.  However, there is no class label available for the GAN distribution samples, and so such comparison isn’t possible.     In contrast, CrossLID only requires availability of class labels for the real distribution samples and hence the oversampling approach can be used.
> Estimation of CrossLID is possible mode-wise, because the estimation is essentially “local”.  One just computes the nearest neighbours of a real sample, using neighbours from the entire GAN distribution.  Then an average is taken over all real samples within a class. In contrast, estimation of FID is essentially “global”, and operates by comparing two distributions. Mode-wise computation of FID may be possible for specific types of GAN models such as conditional GANs. The benefit of CrossLID is that it does not require any such specific GANs for mode-wise performance estimation.  Please also see G2 above.
>
>
> -- "The paper also proposes computing CrossLID in the feature space of a discriminator to make the metric less reliant on an external model. While this is an interesting thing to showcase - FID can also be computed in an arbitrary feature space and the authors do not clarify or investigate whether FID performs similarly.”
>
> RESPONSE 1.5: Please see G2 above.  Although we believe exploring this issue is beyond the scope of our study, we have done some preliminary experiments on FID in a discriminator feature space, and the results indicate that FID can be computed as well using the discriminator activations. Accordingly, we have updated Section 6.1 (“Dependency on external model”) and Table 1 to reflect this finding.
>
> -- "Several experiments get into some unclear value judgements over what the behavior of an ideal metric should be. The authors of FID argue the opposite position of this paper that the metric should be sensitive to low-level changes in addition to high-level semantic content. It is unclear to me as the reader which side to take in this debate.”
>
> RESPONSE 1.6: This is an interesting comment.   We believe that debate over properties of an ideal GAN quality measure is extremely important for the ICLR community.   We do not claim our perspective is the only one, but we do believe it adds an important new dimension to the discussion (and after all, such type of discussion should be one of the purposes of a paper at ICLR).  We have added some extra discussion to “Robustness to small input noise” section of the updated paper about this issue.   See also general comment G3 above.
> A potential drawback of high sensitivity to low noise levels is that the metric may respond inconsistently for images with low noise as compared to images of extremely low quality. For which we show in Fig. 17, FID rates the images with centers occluded by black rectangles to be of better quality than images with 2% Gaussian noise --- quite the opposite to human visual judgment.
>
> -- "I have some final concerns over the fact that the metric is not tested on open problems that GANs still struggle with. Current SOTA GANs can already generate convincing high-fidelity samples on MNIST, SVHN, and CIFAR10. Exclusively testing a new metric for the future of GAN evaluation on the problems of the past does not sit well with me.”
>
> RESPONSE 1.7: We understand it is certainly interesting to test measures on complex datasets.   Our philosophy though, is that evaluation of GAN performance on MNIST/SVHN/CIFAR10 is still far from being a closed issue.  If we have time (i.e. enough computational resources for such a big dataset) during the review response period, we will attempt to evaluate on imagenet, but for feasibility we need to prioritize this suggestion lower compared to the other issues raised by the reviewer(s).
>
> -- "Some questions:
> * Could the authors comment on run time comparisons of the metric with FID/IS?”
>
> RESPONSE: 1.8:  We have added runtime statistics to Appendix I of the paper.  The three measures have increasing running times with a linear trend as sample size increases, but at different pace and scales with CrossLID the lowest and FID the highest.

---

> ### Author Response · Authors · 2018-11-15
> **Response to "Concerns about clarity and scalability of the metric" - Part 1**
>
> Thank you very much for your comments. Please read the above “General Response to Reviewers 1, 2 and 3” for our general response G1, G2 and G3.
>
> -- “Figure 1 is a good quick overview of some of the behaviors of the metric but it is not clear why the MLE estimator of LID should be preferred (or perform any differently) on this toy example from a simple average of 1-NN distances. The same is also appears to be true for the motivating example in Figure 8 as well.
>
> To summarize a bit, I found that the paper did not do the best job motivating and connecting the proposed metric to the task at hand and describing in an accessible fashion its potentially desirable properties.”
>
> RESPONSE 1.1: MLE was identified as the best-performing estimator of LID in Amsaleg et. al. (2018), and for this reason we adopt it here. We have made this more clear when introducing it in Equation 5 (Section 3).  As to why an adaptation of LID would be better than simply averaging 1-NN distances of points, the answer lies in the concentration effect of higher dimensions: as the local intrinsic dimensionality of the data (or, if you prefer, the dimension of the local submanifold) increases, the discriminability of the distance measure diminishes. Unlike methods based on thresholding of neighbor distances (such as Hausdorff distance or other linkage criteria from clustering), LID scores are naturally adaptive to local differences in intrinsic dimensionality. Although the two-dimensional point configurations shown in Fig. 1 are amenable to such techniques, they quickly break down for data in higher dimensions, or across a range of different local intrinsic dimensionalities. Without taking local intrinsic dimensionality into account, we would not know whether a given large 1-NN distance value indicates `large spatial separation' or `conformity within a locality of high intrinsic dimensionality' - implying that the direct use of 1-NN distance information leads to a rather poor assessment of the relationship of a point to its surroundings. The strength of the LID model is that it naturally allows for comparison of local effects across localities of different dimensionality. We have added discussion about these issues to Section 4.1 of the updated paper.
>
> -- “The experimental section performs a variety of comparisons between CrossLID, Inception Score and FID. The general finding of the broader literature that Inception Score has some undesirable properties is confirmed here as well. A potentially strong result showing where CrossLID performs well at inter-class mode dropping, Figure 4, is unfortunately confounded with sample size as it tests FID in a setting using 100x lower than the recommended amount of samples. “
>
>
> RESPONSE 1.2: Actually, in each case the sample size used for calculating FID was 50k.  The  50k samples were created by performing oversampling of  a dataset with n<50k.  E.g. Select 10 classes, each with 100 samples (so a total of n=10*100=1000 instances).   Then create a dataset of size 50k by sampling with replacement 50000 times from this pool of 1000 instances.  We have added a clarifying comment to the “Sensitivity to mode collapse” subsection of Section 6.1.
>
>
> -- "The analysis in this section is primarily in the form of interpretation of visual graphs of the behavior of the metrics as a quantity is changed over different datasets. I have some concerns that design decisions around these graphs (normalizing scales, subtracting baseline values) could substantially change conclusions."
>
> RESPONSE 1.3:  We have added results (graphs with normalized scores) in Appendices G and H of the updated paper.  The conclusions are generally unchanged.   If there are other figures the reviewer would specifically like updated, please let us know.

---

### Author Response · Authors · 2018-11-15
**General Response to Reviewers 1, 2 and 3**

We thank the reviewers for their comments.   Based on the reviews taken together, we first make some general comments (G1-G3), followed by more detailed point by point comments for each review.

G1: We are not arguing CrossLID should “replace” existing measures such as FID.  CrossLID measures GAN quality from a quite different perspective to FID (local rather than global).   It can be deployed as an additional tool for the community to use for understanding and assessing GAN quality, and might be used alongside existing measures like FID or Inception Score.    As an analogy, many measures for validating clustering quality have been developed (both internal and external) - a typical research paper compares different clustering algorithms in terms of their quality.   It is well known that there is no single best quality measure and so it is common for researchers to report performance with respect to several measures, for more robustness of their findings.  Our updated paper now includes mention of these perspectives in the conclusion.

G2: In our study, we have proposed strategies for CrossLID to work effectively through i) using feature space of the DNN to assess distances, and ii) using class labels  to improve learning of certain modes.   Reviewers 1 and 3 argued that such strategies ought also to be tried in conjunction with FID.  We provide more detailed discussion about the feasibility of this below, but at a high level, we would be pleased if our strategies could also help enhance FID (we have updated the paper to note that such strategies might be evaluated for FID).     Such an outcome would add (not detract) value to our overall contribution, since to the best of our knowledge, i) and ii) are novel strategies in the context of assessing GAN quality.

G3: Reviewers 1 and 3 raised the issue of stability.  Is it better for a GAN metric to be sensitive to low levels of noise, or insensitive to low levels of noise?   We have argued that the latter is desirable, whereas the reviewers appear to lean towards the former.   We understand that there is some room for debate here and have updated the paper accordingly.  To the best of our knowledge though, this issue of stability is unaddressed by the GAN community and we pose it as an open question “What level of sensitivity is appropriate for a GAN quality measure applied on images which are highly recognizable, but which contain a low level of noise”?

---

### Author Response · Authors · 2018-11-16
**Major sections updated in the revised paper**

To address the reviewers' comments, the following sections/subsections of the paper was updated:
-Section 4.1 (in the context of CrossLID, in the first paragraph after its definition)
-Appendices F, G, H, I, J, L
We hope the above information would help the reviewers to identify the major changes incorporated in the paper.
Apart from these, sections 3, 6.1, 6.2, and 7 have also been updated with minor changes with respect to reviewers comments.

---

### Meta-Review · Area_Chair1 · 2018-12-14
**Intersting new evaluation metric which might have a scalabilty issue.**

**Confidence:** 4
**Recommendation:** Reject

**Metareview:**

The paper propose a new metric for the evaluation of generative models, which they call CrossLID and which assesses the local intrinsic dimensionality (LID) of input data with respect to neighborhoods within generated samples, i.e. which is based on nearest neighbor distances between samples from the real data distribution and the generator. The paper is clearly written and provides an extensive experimental analysis, that shows that LID is an interesting metric to use in addition to exciting metrics as FID, at least for the case of not to complex image distributions The paper would  be streghten by showing that the metric can also be applied in those more complex settings.